# KNOWLEDGE FUSION OF LARGE LANGUAGE MODELS VIA MODULAR SKILLPACKS

**Guodong Du**[1,2], **Zhuo Li**[1], **Xuanning Zhou**[1], **Junlin Li**[1], **Zesheng Shi**[1], **Wanyu Lin**[2],
**Ho-Kin Tang**[1], **Xiucheng Li**[1], **Fangming Liu**[1,4], **Wenya Wang**[3], **Min Zhang**[1], **Jing Li**[1✉]

[1]Harbin Institute of Technology, Shenzhen, China
[2]The Hong Kong Polytechnic University     [3]Nanyang Technological University
[4]Huazhong University of Science and Technology, China
duguodong7@gmail.com    jingli.phd@hotmail.com

## ABSTRACT

Cross-capability transfer represents a key challenge in large language model (LLM) research, particularly in multi-task integration, model compression, and knowledge fusion. Recent works such as FuseLLM and FuseChat have shown the potential of transferring multiple model capabilities to lightweight models, thereby enhancing adaptability and efficiency. This motivates our investigation into more efficient methods for cross-capability transfer. However, existing model merging approaches primarily focus on homogeneous models, limiting their applicability. For large, heterogeneous models, knowledge distillation with full-parameter fine-tuning often overlooks the student model's inherent capability and risks catastrophic forgetting, while PEFT methods struggle to effectively absorb knowledge from source LLMs. To address these issues, we introduce `GraftLLM`, a novel grafting-based method that stores source model capabilities in a target model + SkillPack format. This approach preserves general capabilities, reduces parameter conflicts, and supports forget-free continual learning and model fusion. We employ a module-aware adaptive compression strategy for parameter updates, ensuring efficient storage while **preserving task-specific knowledge**. The resulting SkillPack serves as a compact and transferable knowledge carrier, ideal for **heterogeneous LLM fusion**. Experiments across various scenarios demonstrate that `GraftLLM` outperforms existing techniques in knowledge transfer, knowledge fusion, and forget-free learning, providing a scalable and efficient solution for cross-capability transfer. The code is publicly available at: https://github.com/duguodong7/GraftLLM.

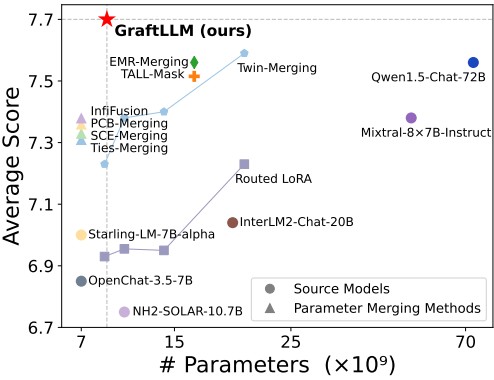
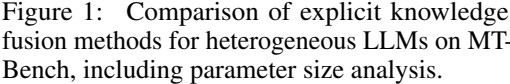

Figure 1: Comparison of explicit knowledge fusion methods for heterogeneous LLMs on MT-Bench, including parameter size analysis.

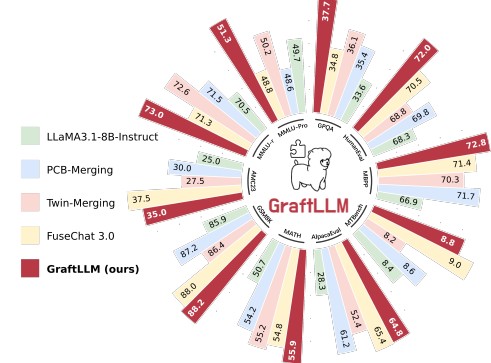

Figure 2: A comprehensive comparison of implicit knowledge fusion methods for heterogeneous LLMs across multiple benchmarks.

---

✉ Corresponding author.

## 1   INTRODUCTION

Cross-capability transfer (Pan et al., 2023; Zhong et al., 2025; Yang et al., 2022; Fujii et al., 2024; Zhao et al., 2024) aims to combine or migrate different skills and task abilities across LLMs, enabling a single model to benefit from capabilities originally distributed among multiple specialized sources. This paradigm has has received increasing attention in LLM research, driving progress in key applications such as multi-task fusion (Yang et al., 2024d), model compression (Wang et al., 2024c; Huang et al., 2024), and continual learning (Tang et al., 2025). KnowPara (Zhong et al., 2024) employs sensitivity-based techniques to extract and align knowledge-specific parameters (Panigrahi et al., 2023) across different models. FuseLLM (Wan et al., 2024a) and FuseChat (Yang et al., 2025b) showcase the potential of distilling multiple models into a lightweight target, while EvoLLM (Akiba et al., 2025) introduces an evolutionary approach (Du et al., 2024c;a) to automatically combine diverse open-source models without extra training data. These methods collectively tackle the core challenge of efficient and reliable knowledge transfer across heterogeneous LLMs.

Existing knowledge grafting methods (Deng et al., 2024) aim to enable cross-capability transfer but are mostly limited to small (Panigrahi et al., 2023; He et al., 2024) or structurally identical models (Yu et al., 2023), which constrains their applicability to heterogeneous LLMs. To address this challenge, we introduce **GraftLLM**, which encodes model capabilities as a combination of a target model and a lightweight *SkillPack*. A SkillPack is a modular, task-specific set of parameter deltas obtained via distillation from heterogeneous source models. This design preserves the strengths of both target and source models, enhances parameter and storage efficiency, mitigates forgetting, and facilitates multi-task transfer and model fusion by reducing parameter conflicts (Yadav et al., 2024). By contrast, knowledge distillation—a widely adopted grafting strategy—typically follows two paradigms: full-parameter distillation and PEFT-based fine-tuning. The former often disregards the student model's intrinsic capabilities and risks catastrophic forgetting (Alexandrov et al., 2024), while the latter, though more parameter-efficient, generally underperforms full fine-tuning and struggles to absorb sufficient task knowledge from source models.

We consider a heterogeneous capability transfer scenario where source model capabilities are extracted via synthetic data (Yang et al., 2025b), integrated into the target model through full-model fine-tuning, and further refined with preference optimization (e.g., DPO (Rafailov et al., 2023)). The resulting parameter deltas capture the specialized knowledge gained during this process. To enable efficient storage and transfer, we introduce a module-aware adaptive compression strategy that compresses these deltas before and after specialization. By adapting pruning (Yu et al., 2023), low-rank decomposition (Lu et al., 2024), and adaptive quantization (Ping et al., 2024) to each module's structure, our method balances compression ratio with task knowledge retention. The compressed representation, termed a *SkillPack*, serves as a compact, transferable knowledge unit, supporting scalable integration and continual specialization without catastrophic forgetting.

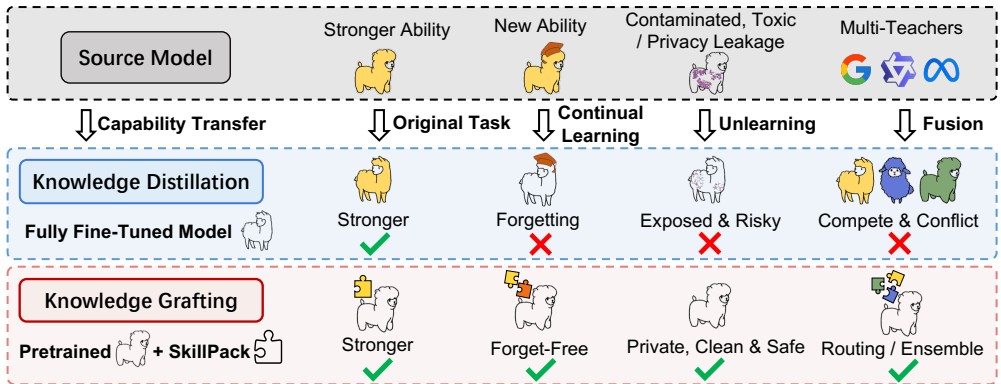

Figure 3:   Comparision of knowledge distillation and knowledge grafting in various scenarios.

We anticipate that the knowledge grafting method will provide advantages in various scenarios, as illustrated in Fig. 3. (1) **First**, it nearly matches full-parameter distillation in learning from a source model with superior original-task capability. (2) **Second**, since grafting does not alter the target model's parameters, it proves highly effective for forget-free learning, allowing the source model to acquire new abilities. (3) **Additionally**, the grafted modules can be easily unloaded, facilitating

unlearning, detoxification, decontamination, and other processes, thus helping mitigate issues like privacy leakage. (4) **Finally**, `GraftLLM` employs a routing mechanism to support model fusion and multi-task learning, avoiding parameter competition and conflict, further enhancing its applicability.

To empirically validate the effectiveness of `GraftLLM`, we conducted extensive experiments in various cross-capability transfer scenarios, demonstrating our approach's advantages from three perspectives: (1) **Knowledge Transfer and Compression**: using LLaMA3 as the target, we grafted capabilities from sources like Qwen-2.5-72B-Instruct (Yang et al., 2024a) under SFT and DPO settings, significantly outperforming PEFT and Twin-Merging (Lu et al., 2024) on general and task-specific tasks. (2) **Knowledge Fusion**:, we tested on 10 benchmarks under both explicit and implicit heterogeneous model fusion scenarios, with LLaMA3.1-8B-Instruct (Dubey et al., 2024a) and Qwen-2.5-7B-Instruct (Yang et al., 2024a) as target models, showing substantial improvements over existing methods, as shown in Fig. 1 and 2. (3) **Forget-free Learning**: our method better mitigates catastrophic forgetting, achieving stronger forget-free learning performance.

This paper makes three significant **contributions**: (1). We highlight the necessity of cross-capability transfer between heterogeneous large language models and identify limitations in existing methods regarding generalization and adaptability. (2). We propose `GraftLLM`, which structures cross-model capabilities as SkillPack, offering high performance, forgetfulness resistance, and easy integration for practical applications. (3). Experiments show `GraftLLM` significantly improves performance in knowledge transfer and compression, heterogeneous model fusion, and forget-free learning tasks.

## 2 RELATED WORK

**Knowledge Distillation**    Knowledge distillation (Hinton et al., 2015) plays a crucial role in enabling capability transfer (Wan et al., 2024a; Zhong et al., 2024; 2025) across heterogeneous large language models (LLMs). Despite the progress made by knowledge distillation methods in merging large language models (LLMs), two main approaches have emerged: one involves complex **multi-task training** (Yang et al., 2025b) for model sharing, but often fails to achieve optimal performance for individual tasks (Shen et al., 2024; Yang et al., 2024c); the other uses **pairwise distillation** (Wan et al., 2024b; Yan et al., 2025) followed by parameter merging (Li et al., 2023a; Matena & Raffel, 2022), but conflicts between tasks during fusion can lead to performance degradation (Yadav et al., 2024). To address this, **routing mechanisms** (Muqeeth et al., 2024; Li et al., 2024a) have been introduced to preserve single-task performance while reducing task interference (Yang et al., 2024e). However, routing requires each branch to be highly parameter-efficient to minimize resource usage (Lu et al., 2024; Kang et al., 2024). While **PEFT methods** such as LoRA (Wu et al., 2024) introduce lightweight adapters, they often fall short of the performance achieved by full-parameter fine-tuning (Ding et al., 2023). To address this, we propose a strategy that first fine-tunes all parameters and then modularizes them, providing stronger support for routing and fusion.

**Model Fusion**    Most existing model merging approaches primarily focus on homogeneous settings, where models share the same pre-trained backbone. Within this scope, Model Grafting (Panigrahi et al., 2023) was first proposed as a technique to transplant a small subset of fine-tuned parameters onto the pre-trained model, effectively recovering the performance of the original fine-tuned model. Meanwhile, Task Arithmetic (Ilharco et al., 2023; Zhang et al., 2023) introduced the concept of task vectors, and Ties-Merging (Yadav et al., 2024) demonstrated the importance of pruning these vectors. Building on this idea, subsequent works like DARE (Yu et al., 2023) and TSV-Merge (Gargiulo et al., 2025) applied it to merging large language models. Beyond task vector pruning, methods such as mask localization (Panigrahi et al., 2023; He et al., 2024), singular value decomposition (SVD) (Wang et al., 2024f; Yuan et al., 2023), and quantization (Frantar et al., 2022; Lin et al., 2024) have also been widely adopted for model compression and merging. For example, Model Tailor (Zhu et al., 2024) generates sparse masks based on salience and sensitivity scores, while Talls Mask (Wang et al., 2024c) and EMR-Merging (Huang et al., 2024) introduce additional masks to localize task-specific information and reduce storage costs. SVD is applied in various contexts: Twin-Merging (Lu et al., 2024) uses it for modular routing, KnOTS (Stoica et al., 2024) for LoRA fusion, and $D^2$-MoE (Gu et al., 2025) for MoE-based LLMs. Methods like BitaDelta (Liu et al., 2024) and Delta-Come (Ping et al., 2024) incorporate quantization for further compression. In our `GraftLLM` work, we propose a module-adaptive delta compression strategy for merging heterogeneous LLM that balances performance and storage efficiency. More comparisons with related work are provided in App. A.

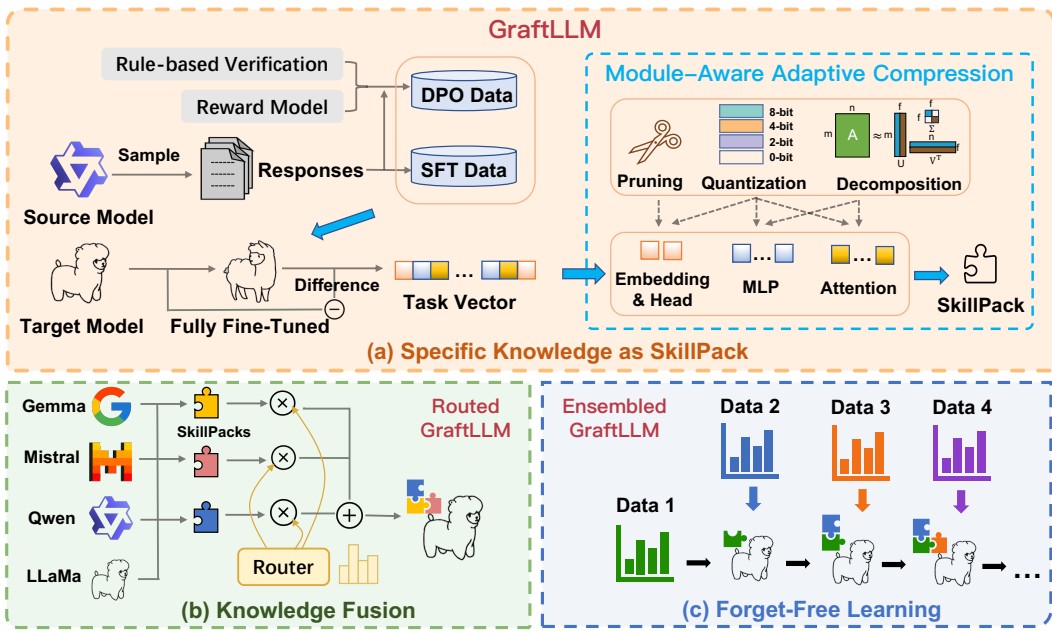

Figure 4: **Overview of GraftLLM**. `GraftLLM` transfers capabilities across heterogeneous LLMs and extracts them into compact **modular SkillPacks**, enabling efficient knowledge fusion.

## 3 METHODOLOGY

In Sec. 3.1, we formalize the problem of efficient LLM fusion. Sec. 3.2 introduces our proposed method, `GraftLLM`, which enables cross-capability transfer between heterogeneous models and encapsulates the acquired knowledge into a compact *SkillPack*. Finally, Sec. 3.3 illustrates how the modularity and composability of *SkillPacks* support downstream applications, including heterogeneous model fusion and forget-free learning.

### 3.1 PROBLEM SETTING

We consider a heterogeneous adaptation scenario involving a source model $\theta_{\text{src}}$ and a target model $\theta_{\text{tgt}}$. To transfer capabilities from the source to the target, we adopt a two-stage training pipeline: supervised fine-tuning (SFT) followed by direct preference optimization (DPO).

The parameters after this two-stage adaptation are denoted $\theta_{\text{tgt}}^{\star}$, and we define the difference from the original parameters as the **delta parameters**:

$$\Delta\theta = \theta_{\text{tgt}}^{\star} - \theta_{\text{tgt}}, \tag{1}$$

which captures the task-specific adaptation knowledge and serves as the foundation for subsequent modular compression and transfer.

To enable efficient storage and transfer, we compress $\Delta\theta$ using a **module-specific adaptive strategy**. Each submodule $m \in \mathcal{M}$ is compressed with a dedicated operator $C_m(\cdot)$, selected based on its functional role and sensitivity. The compression may involve pruning, low-rank decomposition, or quantization, with bitwidth adaptively assigned according to the importance of each component. The resulting compressed update is:

$$\widehat{\Delta\theta} = \{C_m(\Delta\theta_m)\}_{m\in\mathcal{M}}, \tag{2}$$

which forms a *SkillPack*—a compact, transferable representation of the acquired task knowledge, suitable for heterogeneous model fusion, as shown in Fig. 4.

### 3.2 KNOWLEDGE AS A SKILLPACK

To achieve compact and transferable skill representations, we propose a **module-aware adaptive compression strategy**, which—unlike previous uniform compression methods—applies different operations based on each module's role, sensitivity, and compression difficulty. As

shown in Fig. 5 and Fig. 6, moderate pruning preserves performance for the **Embedding** and **Output Head**. For **Attention** modules, the fast-decaying singular value spectrum allows low-rank SVD to compress projection matrices without significantly reducing representational capacity. **MLP** modules, with strong nonlinear transformations, require conservative compression to retain critical singular vectors and avoid performance degradation. Accordingly, we assign module-specific compression operators to the delta parameters as follows:

- **Embedding and Output Head.** We apply magnitude pruning with a retention ratio $\alpha$, preserving the weights with the top $\alpha$ proportion of absolute magnitudes:

$$\Delta\theta^{\text{embed}} = \text{Prune}_\alpha(\Delta\theta^{\text{embed}}). \qquad (3)$$

- **Attention Modules.** For attention blocks, we apply low-rank decomposition using SVD:

$$\Delta\theta^{\text{mlp}} \approx \mathbf{U}\Sigma\mathbf{V}^\top, \quad \text{s.t. } \text{rank}(\Sigma) = r \qquad (4)$$

where $\Delta\theta^{\text{mlp}} \in \mathbb{R}^{h_{\text{out}}\times h_{\text{in}}}$, $\mathbf{U}_r \in \mathbb{R}^{h_{\text{out}}\times r}$, $\Sigma_r \in \mathbb{R}^{r\times r}$, and $\mathbf{V}_r \in \mathbb{R}^{h_{\text{in}}\times r}$ correspond to the top $r$ components.

- **MLP Modules.** For MLP modules, we employ a conservative SVD scheme that retains essential ranks, with the truncation rank determined by the cumulative explained variance under an energy threshold $\beta$:

$$\sum_{i=1}^{k} \sigma_i^2 = \beta \sum_{i=1}^{\min(d_{\text{out}}, d_{\text{in}})} \sigma_i^2, \qquad (5)$$

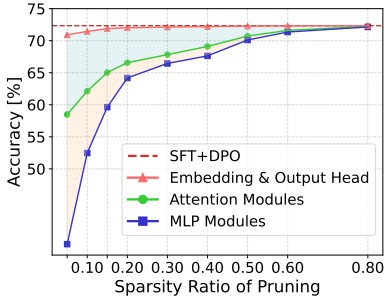

Figure 5: Performance of delta parameters across modules under different pruning ratios.

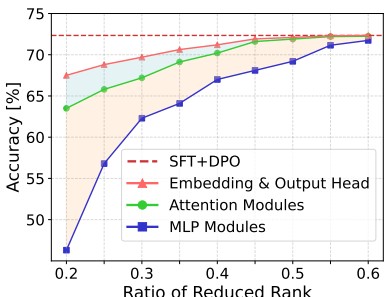

Figure 6: Performance of delta parameters across modules under different reduced rank ratios.

To further reduce storage overhead, we apply mixed-precision quantization to the pruned matrix or SVD-derived components. Each SVD component is quantized with a bit precision $k$, adaptively chosen based on its importance in the decomposition.

$$\hat{\theta} = \text{Quant}_k(\theta, \mathbf{x}) = \underset{\hat{\theta}}{\arg\min} \|\theta\mathbf{x} - \hat{\theta}\mathbf{x}\|^2, \qquad (6)$$

where $\text{Quant}_k$ denotes a $k$-bit quantization operator ($k > 1$). For each group of singular vectors indexed by $[r] = r_{\text{begin}} : r_{\text{end}}$, we apply group-wise quantization with GPTQ (Frantar et al., 2022) as follows:

$$\hat{\mathbf{V}}_{[r]}^\top = \text{Quant}_k\left(\mathbf{V}_{[r]}^\top, \mathbf{x}\right), \hat{\mathbf{U}}_{[r]} \quad = \text{Quant}_k\left(\mathbf{U}_{[r]}, \Sigma_{[r]} \cdot \hat{\mathbf{V}}_{[r]}^\top \cdot \mathbf{x}\right), \qquad (7)$$

where $\Sigma_{[r]}$ denotes the diagonal matrix of singular values corresponding to the selected rank range. The quantization precision $k$ can be adaptively adjusted across different groups based on the relative importance of singular values.

### 3.3 SKILLPACK COMPOSITION AND ROUTER MECHANISM

GraftLLM enables modular and composable integration of task-specific knowledge across heterogeneous LLMs through **SkillPacks** $\widehat{\Delta\theta}$. Each SkillPack is first **decoded through dequantization** to obtain $\Delta\theta^{(dq)}$, and then **reconstructed via truncated SVD** to recover the task-specific delta parameters:

$$\Delta\theta^{(dq)} \approx U\Sigma V^\top = \Delta\theta, \qquad (8)$$

where $U, \Sigma, V$ are obtained from the truncated SVD decomposition. The reconstructed delta $\Delta\theta$ is then **added back to the base model parameters** to produce the final fused model:

$$\theta_{\text{fused}} = \theta_{\text{tgt}} + \Delta\theta. \qquad (9)$$

To support flexible and selective integration across tasks, a **router function** $\mathcal{R}$ is introduced. The router determines which SkillPack is applied to which submodule or task-specific region of the target model. For example, for a set of $n$ SkillPacks $\{\widehat{\Delta\theta}_i\}_{i=1}^n$, the fused model is computed as:

$$\theta_{\text{fused}} = \theta_{\text{tgt}} + \sum_{i=1}^n \mathcal{R}(\widehat{\Delta\theta}_i), \tag{10}$$

where $\mathcal{R}$ can be instantiated in two ways:

- **Classifier-based router**: a lightweight feed-forward network trained to predict the most suitable source model or SkillPack based on input features.
- **Manual task-type assignment**: a deterministic mapping from known task types to their corresponding SkillPacks.

During inference, we typically use **top-1 routing** for efficiency, ensuring that only the most relevant SkillPack is activated. This strategy minimizes inference overhead while preserving the benefits of modular, task-specific delta parameters. More details provided in App. B.3

Overall, this formulation provides a unified interface for modular knowledge transfer, enabling both **heterogeneous model fusion** and **task-adaptive capability integration** in a principled, efficient, and scalable manner.

## 4   EXPERIMENTAL SETUP

### 4.1   BASELINE METHODS

For **pairwise LLM grafting**, we evaluate two categories of baselines: (1) **PEFT methods**, comparing LoRA under varying rank settings in both SFT and DPO stages; (2) **Task Vector Compression**, which evaluates full-parameter tuning followed by magnitude pruning (Yu et al., 2023; Yadav et al., 2024), SVD (Lu et al., 2024; Stoica et al., 2024), or quantization (Ping et al., 2024; Yang et al., 2025a) across varying compression ratios.

For **heterogeneous knowledge fusion**, we benchmark against: (1) **Multi-teacher distillation** (e.g., FuseLLM (Wan et al., 2024a)); (2) **Parameter merging** approaches such as Task Arithmetic (Ilharco et al., 2023), TIES-Merging (Yadav et al., 2024), SCE-Merging (Wan et al., 2024b), PCB-Merging (Du et al., 2024b), DARE (Yu et al., 2023), and InfiFusion (Yan et al., 2025); (3) **Routing-based** methods, including Routed LoRA (Hu et al., 2022) and Twin-Merging (Lu et al., 2024), and (4) **Mask-based fusion** strategies like TALL Mask (Wang et al., 2024c) and EMR-Merging (Huang et al., 2024), which leverage unified task vectors and localization.

For **forget-free learning**, we use LoRA, Model Grafting (Panigrahi et al., 2023), and Model Tailor (Zhu et al., 2024) as baselines. Details of all baselines are provided in App. D.

### 4.2   DATASETS AND ARCHITECTURES

To showcase the effectiveness of GraftLLM, we conduct a comprehensive evaluation across multiple domains, including instruction following, question answering, reasoning, mathematics, and coding. We use 10 established benchmarks, grouped into four categories, with domain-specific response sampling strategies to ensure fair comparison. Full benchmark details are available in App. E.3.

For pairwise LLM grafting in Sec. 5.1 and Fig. 7, 8, we uses Llama-3.1-8B-Instruct as the target model, grafting capabilities from strong source model Qwen-2.5-72B-Instruct (Yang et al., 2024a). For explicit knowledge fusion in Sec. 5.2 and Tab. 1, we follow the FuseChat 2.0 (Wan et al., 2024b) setup by fusing chat-centric LLMs of varying architectures and scales, using OpenChat-3.5-7B (Wang et al., 2024b) as the pivot model and six representative chat models as sources. For implicit fusion in Sec. 5.2 and Tab. 2, we adopt the FuseChat 3.0 (Yang et al., 2025b) setup with Llama-3.1-8B-Instruct and Qwen-2.5-7B-Instruct as target models and 4 different stronger LLMs. For forget-free learning in Sec. 5.3 and Tab. 3, we sequentially acquire math and coding abilities using both SFT and DPO datasets. Further architectural details are provided in App. E, while additional implementation details can be found in App. F, covering training procedures F.1, hyperparameter settings F.2, and computational resources and runtimes F.3.

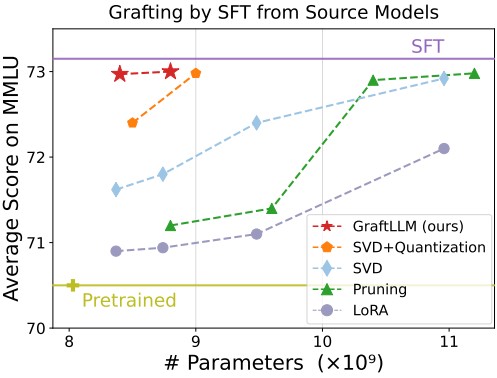
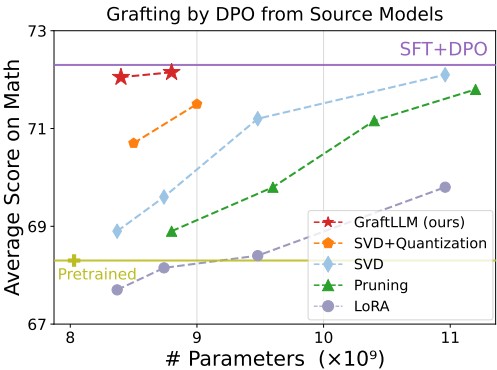

Figure 7: Comparison of parameter efficiency and MMLU performance across different methods for LLM capability transfer with SFT.

Figure 8: Comparison of parameter efficiency and average performance on GSM8K and MATH across different methods under the DPO setting.

## 5   RESULTS

In this section, we evaluate `GraftLLM` in various settings, comparing it with other methods, including pairwise heterogeneous LLM grafting 5.1, knowledge fusion 5.2, and forget-free learning 5.3, while also highlighting its potential for unlearning tasks like model detoxification (see App. C.2).

### 5.1   PAIRWISE GRAFTLLM

We demonstrate the effectiveness of parameter-efficient capability transfer between paired models. Fig. 7 and 8 show that while PEFT and other compression methods perform reasonably well in simple SFT scenarios, their effectiveness drops significantly—or even fails—under more complex DPO settings. In contrast, our method consistently achieves performance close to a fully fine-tuned target model, highlighting its robustness and efficiency.

### 5.2   GRAFTLLM FOR KNOWLEDGE FUSION

We explore two approaches to LLM knowledge fusion: explicit fusion, which aligns tokens and probability distributions, and implicit fusion, which transfers knowledge through generated data.

**Explicit Knowledge Fusion.**    As shown in Tab. 1, compared to the best results from Merging-based LLM fusion, our approach achieves a significant performance boost without introducing a large number of additional parameters. Compared to routing-based fusion methods, our approach achieves better performance with lower parameter cost. Unlike Twin Merging, which relies on higher ranks, our method delivers superior results more efficiently. Compared to TALL-Mask and EMR-Merging, we avoid the overhead introduced by using a unified task vector.

Compared to the source models, our approach improves the target model, OpenChat-3.5-7B, with only a 28% increase in parameter size, achieving performance comparable to Mixtral-8x7B-Instruct and Qwen1.5-Chat-72B. In fact, on MT-Bench, our model outperforms all source models, setting a new benchmark. Additionally, on AlpacaEval 2.0, it shows an 8.07% improvement over the best parameter fusion method. More result deails are in App. C.2.

**Implicit Knowledge Fusion.**    We evaluate the effectiveness of implicit heterogeneous model fusion on 10 benchmark tasks, as shown in Tab. 2, comparing three representative methods. (1) PCB-Merging (pairwise distillation + parameter fusion) distills knowledge from multiple models and merges their parameters, but suffers from conflicts between source models, limiting its ability to balance multi-task performance. (2) Twin-Merging (pairwise distillation + routing) uses model decomposition for routing-based fusion, but experiences significant performance loss during decomposition, resulting in the weakest performance overall. (3) FuseChat-3 (multi-teacher distillation) integrates knowledge from multiple tasks, yet still falls short of task-specific upper bounds—especially on the GPQA (Rein

Table 1: Overall results of explicit LLM knowledge fusion on AlpacaEval 2.0 and MT-Bench. The best-performing results for both parameter merging and routing-based methods are shown in **bold**, while the performance difference between the two is highlighted in **green**.

| Model | #Params | AlpacaEval 2.0 (GPT-4-1106-Preview) | | MT-Bench (GPT-4-0125-Preview) | | |
|---|---|---|---|---|---|---|
| | | Win Rate | LC Win Rate | 1st Turn | 2nd Turn | Average Score |
| **Source LLMs** | | | | | | |
| OpenChat-3.5-7B Wang et al. (2024a) | 7B | 10.20 | 14.90 | 7.14 | 6.55 | 6.84 |
| Starling-LM-7B-alpha Zhu et al. (2023) | 7B | 14.20 | 14.70 | 7.54 | 6.49 | 7.01 |
| NH2-SOLAR-10.7B Kim et al. (2023) | 10.7B | 12.22 | 18.13 | 7.11 | 6.36 | 6.74 |
| InternLM2-Chat-20B Cai et al. (2024) | 20B | 21.70 | 18.70 | 7.78 | 6.34 | 7.06 |
| Mixtral-8x7B-Instruct Jiang et al. (2024) | 8x7B | 18.30 | 23.70 | 7.76 | 7.00 | 7.38 |
| Qwen1.5-Chat-72B Bai et al. (2023) | 72B | 26.50 | 36.60 | 7.83 | 7.36 | 7.59 |
| **Multi-teacher Distillation** | | | | | | |
| FuseLLM[ICLR24] Wan et al. (2024a) | 7B | 10.56 | 14.50 | 7.36 | 6.40 | 6.88 |
| **Pairwise Distillation + Parameter Merging** | | | | | | |
| Task Arithmetic[ICLR23] Ilharco et al. (2023) | 7B | 10.67 | 15.78 | 7.54 | 6.78 | 7.22 |
| Ties-Merging[NeurIPS23] Yadav et al. (2024) | 7B | 11.55 | 16.73 | 7.59 | 7.03 | 7.31 |
| SCE-Merging[arXiv24] Wan et al. (2024b) | 7B | 11.63 | 16.89 | 7.61 | 7.05 | 7.33 |
| PCB-Merging[NeurIPS24] Du et al. (2024b) | 7B | 11.82 | 17.22 | 7.71 | 7.01 | 7.36 |
| PCB-Merging+DARE[ICML24] Yu et al. (2023) | 7B | **11.96** | **17.35** | **7.79** | 6.99 | **7.39** |
| InfiFusion[arXiv25] Yan et al. (2025) | 7B | 11.74 | 17.21 | 7.68 | **7.08** | 7.38 |
| **Pairwise Distillation + Router** | | | | | | |
| Routed LoRA r512 | 14.1B | 10.16 | 15.48 | 7.14 | 6.75 | 6.95 |
| Routed LoRA r1024 | 21B | 12.57 | 19.41 | 7.52 | 6.92 | 7.23 |
| TALL-Mask[ICML24] Wang et al. (2024c) | 16.7B | 13.69 | 22.76 | 7.92 | 7.14 | 7.53 |
| EMR-Merging[NeurIPS24] Huang et al. (2024) | 16.7B | 14.52 | 23.10 | 7.96 | 7.15 | 7.56 |
| Twin-Merging r512 | 14.1B | 12.20 | 19.90 | 7.74 | 7.07 | 7.40 |
| Twin-Merging r1024 | 21B | 15.93 | 24.81 | 8.01 | 7.18 | 7.59 |
| **Routed GraftLLM (Ours)** | 9.2B | **16.56**(+4.6) | **25.42**(+8.07) | **8.05**(+0.26) | **7.35**(+0.27) | **7.70**(+0.31) |

Table 2: Overall results of implicit LLM knowledge fusion across 10 benchmark tasks.

| Category | Benchmark | Llama-3.1-8B-Instruct | | | | | Qwen-2.5-7B-Instruct | | | | |
|---|---|---|---|---|---|---|---|---|---|---|---|
| | | Base | PCB-Merging | Twin-Merging | Fuse Chat-3 | **Routed GraftLLM** | Base | PCB-Merging | Twin-Merging | Fuse Chat-3 | **Routed GraftLLM** |
| General | MMLU-Pro | 49.7 | 48.6 | 50.2 | 48.8 | **51.3** | 54.0 | 53.7 | 54.5 | 52.8 | **55.4** |
| | MMLU-redux | 70.5 | 71.5 | 72.6 | 71.3 | **73.0** | 75.1 | 75.3 | 74.8 | 74.6 | **76.2** |
| | GPQA-Diamond | 33.6 | 35.4 | 36.1 | 34.8 | **37.7** | 34.7 | 34.2 | 36.8 | 33.9 | **38.1** |
| Mathematics | GSM8K (0 shot, CoT) | 85.9 | 87.2 | 86.4 | 88.0 | **88.2** | 91.7 | 91.5 | 91.3 | 91.7 | **92.0** |
| | MATH (0 shot, CoT) | 50.7 | 54.2 | 55.2 | 54.8 | **55.9** | 75.0 | 73.2 | 72.1 | 73.5 | 75.0 |
| | AMC 23 (0 shot, CoT) | 25.0 | 30.0 | 27.5 | 37.5 | 35.0 | 52.5 | 52.5 | 50.0 | 57.5 | 55.0 |
| Coding | HumanEval (0 shot) | 68.3 | 69.8 | 68.8 | 70.5 | **72.0** | 85.4 | 83.1 | 81.9 | 79.9 | **85.6** |
| | MBPP (0 shot) | 66.9 | 71.7 | 70.3 | 71.4 | **72.8** | 80.2 | 82.7 | 81.6 | 83.1 | **84.5** |
| Instruction Following | AlpacaEval-2 (LC %) | 28.3 | 61.2 | 52.4 | **65.4** | 64.8 | 34.2 | 58.9 | 53.3 | 63.6 | 61.5 |
| | MT-Bench | 8.4 | 8.6 | 8.2 | **9.0** | 8.8 | 8.4 | 8.6 | 7.8 | 9.0 | 8.7 |
| | Average | 48.7 | 53.8 | 52.8 | 55.2 | **56.0** | 59.0 | 61.4 | 60.4 | 62.0 | **63.2** |

et al., 2023) benchmark, where other tasks offer little benefit. In contrast, our method combines the performance strengths of pairwise distillation with the parameter efficiency of modular routing, effectively reducing task conflicts and fusion costs. When using LLaMA3.1-8B-Instruct and Qwen-2.5-7B-Instruct as target models, our approach achieves average performance gains of 0.8 and 1.2, respectively, demonstrating significant advantages. More result deails are in App. C.1.

## 5.3   GRAFTLLM FOR FORGET-FREE LEARNING

Table 3: Forget-free learning results on code and math tasks using LLaMA3.1-8B-Instruct.

| Method | Additional Parameters | Code Benchmarks (Original task) | | | | Math Benchmarks (New task) | | | Average |
|---|---|---|---|---|---|---|---|---|---|
| | | HumanEval | HumanEval+ | MBPP | MBPP+ | GSM8K | MATH | AMC23 | |
| LLaMA3.1-8B-Instruct | - | 68.3 | 61.6 | 66.9 | 54.8 | 85.9 | 50.7 | 25.0 | 59.0 |
| Multi LoRA r256 | 1.48B | 68.8 | 61.8 | 67.7 | 55.6 | 86.2 | 51.3 | 25.0 | 59.5 |
| Model Grafting[ICML23] | 803M | 70.4 | 63.9 | 69.1 | 57.5 | 87.2 | 53.4 | 27.5 | 61.3 |
| Model Tailor[ICML24] | 803M | 71.4 | 64.2 | 71.1 | 59.4 | 87.6 | 54.5 | 27.5 | 62.2 |
| **GraftLLM (ours)** | 803M | **72.0** | **65.2** | **72.2** | **61.8** | **88.2** | **55.9** | **35.0** | **64.3** |

We evaluate `GraftLLM` in a forget-free learning setting, where `LLaMA3.1-8B-Instruct` is first trained on code (original task) and then on math (new task), using data generated from stronger source models. The final model is evaluated on seven benchmarks in total—four for code and three for math. Under the same 10% parameter budget as prior methods like Model Grafting and Model Tailor, `GraftLLM` delivers consistently stronger performance while mitigating forgetting, outperforming existing approaches by an average of 2.1% (Tab. 3). More deails are in App. C.1.

## 5.4    Performance on Highly Distinct Fusion Domains

Our method is explicitly designed to decouple conflicting task behaviors into separate SkillPacks, preventing cross-task interference and enabling near-lossless capability fusion—even when the underlying tasks are highly distinct. To validate this design in scenarios with highly divergent domains, we conducted a new experiment involving finance, law, and biomedicine, following an experimental setup inspired by AdaptLLM (Cheng et al., 2023). We first report the performance of models fine-tuned individually on each domain. As shown in the table below, a model fine-tuned on one domain suffers substantial degradation on other domains, sometimes performing worse than the base model, highlighting the limitations of traditional merging approaches in handling conflicting updates.

In contrast, SkillPack-based fusion effectively isolates domain-specific delta parameters and recombines them with minimal interference, achieving near-lossless multi-domain performance. Furthermore, even when compressing the model to just 10% of its original parameters, our method still reaches nearly 99% of the original performance. This means that the performance originally requiring three separate 7B fine-tuned models can now be matched by GraftLLM with only an additional 30% of parameters, demonstrating significant parameter efficiency without sacrificing accuracy.

Table 4: Performance of GraftLLM compared with baselines across Biomedicine, Finance, and Law domains. The average column also indicates relative performance to the reference.

| Methods | Params | Biomedicine | Finance | Law | Average |
|---|---|---|---|---|---|
| LLaMA-7B | 7B | 44.2 | 58.6 | 34.2 | 45.7 |
| LLaMA-7B-Bio | 7B | 47.3 | 57.9 | 34.5 | 46.6 |
| LLaMA-7B-Finace | 7B | 43.7 | 63.4 | 34.0 | 57.0 |
| LLaMA-7B-Law | 7B | 44.1 | 58.2 | 38.5 | 46.9 |
| AdaptLLM-7B | 3x7B | 47.3 | 63.4 | 38.5 | $49.7_{100\%}$ |
| FuseChat[ICLR 25] | 7B | 45.6 | 60.1 | 36.3 | $47.3_{95\%}$ |
| Twin-Merging[NeurIPS 24] | 15.3B | 45.9 | 61.3 | 36.7 | $47.9_{96\%}$ |
| **GraftLLM(Ours)** | 9.1B | **47.2** | **63.4** | **38.2** | $\mathbf{49.6}_{99\%}$ |

## 6    Analysis

### 6.1    Ablation Study

We conduct an ablation study to assess the impact of each component within module-aware adaptive strategy. As shown in Tab. 5, we evaluate the effect of replacing or removing individual compression modules, using the LLaMA3.1-8B-Instruct model on the GSM8K and MATH validation sets. To ensure fairness, all configurations maintain a comparable overall compression ratio of approximately 5%. The results reveal that different model components benefit from tailored compression strategies. Quantization emerges as a critical factor for preserving performance, with mixed-precision quantization causing minimal degradation. In contrast, the MLP modules exhibit high sensitivity to compression: applying suboptimal methods to these layers leads to notable performance drops, underscoring their importance in the compressed model architecture. More details of the ablation study are provided in App. B.1.

Table 5: Ablation study on module-aware adaptive strategy on the MATH task.

| Methods | Null | w/o Quantization | w/o Mixed Quant | Pruning | SVD | Low Rank SVD. |
|---|---|---|---|---|---|---|
| Embedding and Output Head | 71.3 | 71.8 | 71.8 | **72.1** | 71.9 | 71.7 |
| MLP Modules | 68.7 | 69.2 | 71.5 | 70.2 | **72.1** | 71.2 |
| Attention Modules | 70.7 | 71.3 | 71.8 | 71.2 | 71.8 | **72.1** |

## 6.2 EFFECT OF TASK DIFFICULTY AND DATA SETTINGS

We assess our method across diverse settings, including varying sample sizes, numbers of source models, and task difficulties. We also study the impact of different compression ratios under both SFT and DPO paradigms. As shown in Fig. 9a and 9b, with a compression ratio (CR) 10%, our method consistently retains nearly 100% of the original performance in both SFT and DPO settings. In a CR ratio 5%, performance decreases as task difficulty increases - particularly under DPO - highlighting the greater challenge of compression in preference-aligned scenarios. Fig. 9c shows our method is robust on simpler tasks but less stable on DPO-based instruction-following tasks, highlighting both strengths and limitations across alignment challenges. Further analysis of compression hyperparameters, including rank and mixed-precision ratios, is in App. B.2.

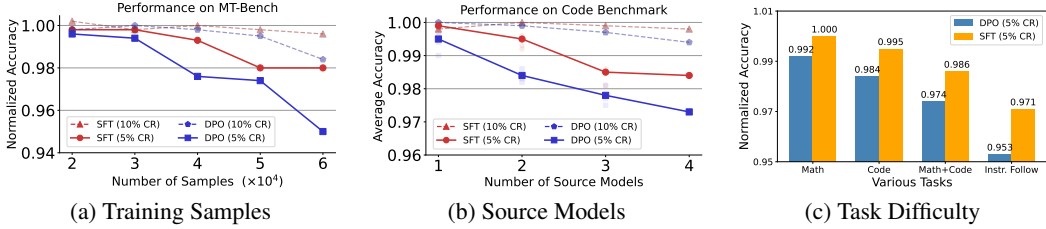

(a) Training Samples  (b) Source Models  (c) Task Difficulty

Figure 9: Performance trends across task difficulty using LLaMA3.1-8B-Instruct.

## 6.3 ROUTER BEHAVIOR AND SKILLPACK INTERACTION

We further examine how the router behaves under different task relationships and evaluate whether combining multiple SkillPacks can provide additional benefits. When SkillPacks correspond to highly similar tasks(e.g., different Chat LLMs), their latent features naturally overlap, making the distinction between experts less pronounced. In these cases, top-1 routing is inherently ambiguous, and ensembling the outputs of multiple SkillPacks—weighted by the router—can yield noticeable performance gains, although with increased inference cost.

In contrast, when SkillPacks correspond to clearly distinct domains such as finance and biomedicine, the router learns a confident mapping to the most relevant expert. Cross-expert ensembling brings little benefit because their knowledge does not reinforce each other. This matches our empirical findings. Overall, the router leverages complementary skills for aligned tasks, avoids interference for divergent ones, and preserves efficiency through selective activation.

Table 6: Analysis of router behavior and multi-SkillPack ensembling across three fusion settings.

| Methods | Explicit LLM Fusion (AlpacaEval 2.0) | Implicit LLM Fusion (Avg. of 10 benchmarks) | Finance+Law+Bio LLM Fusion | Inference Overhead |
|---|---|---|---|---|
| FuseChat | 11.63 / 16.89 | 55.2 | 47.33 | 1x |
| Grafting Top 1 SkillPack | 16.56 (**+4.9**) / 25.42 (**+8.5**) | 56.0 (**+0.8**) | 49.62 (**+0.23**) | 1.03x |
| Grafting Top 2 SkillPacks | 17.33 (+0.8) / 26.28 (+0.9) | 56.3 (+0.3) | **49.64** (+0.02) | 2.03x |
| Grafting Top 3 SkillPacks | **17.49** (+0.2) / **26.72** (+0.4) | **56.4** (+0.1) | 49.62 (-0.01) | 3.03x |

## 6.4 LIMITATION AND FUTURE WORK

While our approach provides insights into knowledge transfer between LLMs, it relies on the quality of prior supervised fine-tuning (SFT) and Direct Preference Optimization (DPO), with suboptimal distillation limiting its ability to fully capture source model capabilities. Future work may explore alternative inference strategies and develop automated, robust methods for compression operations during deployment to improve efficiency, scalability, and robustness.

## 7 CONCLUSIONS

We present `GraftLLM`, a scalable framework for efficient cross-capability transfer in large language models. By compressing task-specific updates into modular SkillPacks, our method preserves knowledge while avoiding interference and forgetting. Experiments show strong performance in knowledge fusion and continual learning, outperforming prior methods under various settings.

ETHICS STATEMENT

This research was conducted in full accordance with established ethical standards in artificial intelligence and machine learning. All experiments utilized publicly available datasets and models under their respective licenses, without involving any personally identifiable or sensitive information. The proposed methods are intended strictly for academic and scientific purposes, aiming to advance understanding in machine learning rather than for deployment in high-stakes decision-making without appropriate safeguards.

We acknowledge that advances in AI systems can entail potential societal risks, including concerns related to fairness, misuse, privacy, and environmental impact arising from computational resource demands. To address these issues, we prioritize responsible reporting of results, transparent disclosure of limitations, and a clear distinction between research contributions and downstream applications.

Future research building upon this work should continue to evaluate potential ethical implications—particularly regarding bias, safety, and dual-use risks—and implement appropriate measures to promote beneficial, equitable, and responsible outcomes.

REPRODUCIBILITY STATEMENT

Our implementation, including all code, training scripts, and evaluation datasets, is available at: https://github.com/duguodong7/GraftLLM

ACKNOWLEDGEMENTS

This work was supported in part by National Natural Science Foundation of China (62476070), Shenzhen Science and Technology Program (JCYJ20241202123503005, GXWD20231128103232001, ZDSYS20230626091203008, KQTD20240729102154066), Department of Science and Technology of Guangdong (2024A1515011540) and National Key R&D Program of China (SQ2024YFE0200592). This work was also supported in part by the Major Key Project of PCL under Grant PCL2025A10 and PCL2024A06, and in part by the Shenzhen Science and Technology Program under Grant RCJC20231211085918010.

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

APPENDIX

# Appendix for Knowledge Fusion of LLMs via Modular SkillPacks

## OVERVIEW

This paper proposes a knowledge grafting approach that efficiently transfers capabilities from heterogeneous LLMs to a target LLM using modular *SkillPacks*. The appendix is structured according to our key contributions. We also make the project code available via an anonymous link for reproducibility: https://anonymous.4open.science/r/GraftLLM-6DaGDCda326B

- Appendix A (Novelty and Contribution) provides additional experimental results on knowledge compression as well as task-level results from the knowledge fusion experiments.
- Appendix B (Additional Analysis) includes ablation studies, hyperparameter analysis, and time cost evaluation for the search process.
- Appendix C (Additional Results) outlines the computational resources and runtimes, along with the training details and evaluation metrics.
- Appendix D (Baselines details) provides a detailed baseline description.
- Appendix E (Datasets details) provides a detailed dataset description.
- Appendix F (Implementation details) provides a detailed dataset description.

## A   NOVELTY AND CONTRIBUTION

We underscore the importance of cross-capability transfer across heterogeneous LLMs and identify key limitations in current methods regarding generalization and adaptability. To this end, we propose GraftLLM, which encapsulates transferable skills as compact *SkillPacks*, offering high performance, robustness to forgetting, and practical integrability. To clearly demonstrate the innovation of our method, we conduct a comparative analysis with existing state-of-the-art baseline methods.

**Comparison with Multi-Teacher Distillation.**    (Wan et al., 2024a; Yang et al., 2024f; Zhang & Yang, 2021) Our method offers several advantages:

1. It avoids the complex training procedures required by multi-task learning.
2. It achieves higher single-task performance ceilings.
3. It is naturally suited for distributed training and federated learning scenarios.

**Comparison with Pairwise Distillation + Parameter Merging.**    Compared to approaches that directly merge parameters after distillation (Du et al., 2024b; 2025a; Yan et al., 2025), our method:

1. Employs a routing mechanism to avoid conflicts between capabilities from different source models.
2. Circumvents the challenge of merging parameters with large differences.
3. Ensures balanced parameter allocation across tasks to mitigate interference.

**Comparison with Pairwise Distillation + Router.**

1. Compared to methods like TALL-Mask (Wang et al., 2024c) and EMR-Merging (Huang et al., 2024) that rely on unified task vectors, our method achieves superior parameter efficiency.
2. Compared to Twin-Merging (Lu et al., 2024), our approach supports large language models and preserves near full-task performance.

**Comparison with PEFT-based distillation + Router.**    Our method provides significantly stronger performance by first applying full-parameter fine-tuning to fully extract the capabilities of the source model, and then using compression to reduce storage overhead. This enables our approach to preserve critical task knowledge that PEFT-based methods (e.g., LoRa-MoE) (Ding et al., 2023; Hu et al., 2022; Du et al., 2025b) often fail to capture, especially in complex scenarios such as Direct Preference Optimization (DPO), where lightweight adapters struggle to inherit nuanced decision boundaries and preference reasoning from the teacher model.

**Comparison with Delta Compression Methods.**    We introduces several key improvements:

1. We propose a **module-adaptive delta compression strategy**, which prioritizes and compresses parameter updates based on the functional significance of each module—an aspect not considered in previous works (Liu et al., 2024; Ping et al., 2024; Yang et al., 2025a).

2. Previous compression schemes typically neglect challenging tasks such as DPO, limiting their robustness across diverse applications.

3. Our method is specifically designed for **source-model capability transfer**, whereas prior techniques target more general or different objectives.

4. We explicitly consider downstream usability in scenarios like **knowledge fusion and forget-free learning**, improving long-term flexibility.

5. Our method achieves a better trade-off between **performance and storage efficiency** compared to earlier approaches.

# B  ADDITIONAL ANALYSIS

## B.1  ADDITIONAL ABLATION STUDIES

To verify the generality of our conclusions, we conduct an additional ablation study on the HumanEval Plus benchmark, using the same `LLaMA3.1-8B-Instruct` model and keeping the overall compression ratio at approximately 5% across all settings, as discussed in Section Analysis 6.1. Similar to the results on math task, we examine the effect of replacing or removing individual compression modules. The results in Table 7 show consistent patterns: Mixed-precision quantization continues to be the key to maintaining performance, while MLP modules remain highly sensitive to compression. Applying suboptimal strategies to MLP layers results in clear performance drops, again highlighting the importance of our module-aware adaptive strategy across different tasks.

Table 7: Ablation study on module-aware adaptive strategy on the HumanEval Plus benchmark.

| Methods | Null | w/o Quantization | w/o Mixed Quant | Pruning | SVD | Low Rank SVD. |
|---|---|---|---|---|---|---|
| Embedding and Output Head | 64.3 | 64.8 | 64.8 | **65.2** | 64.9 | 64.4 |
| MLP Modules | 62.7 | 63.8 | 64.8 | 63.5 | **65.2** | 64.4 |
| Attention Modules | 63.2 | 64.7 | 64.9 | 64.1 | 64.4 | **65.2** |

## B.2  ADDITIONAL HYPERPARAMETERS ANALYSIS

We further analyze the impact of compression hyperparameters—including the SVD decomposition rank and mixed-precision quantization ratios—on model performance. Experiments are conducted on the LLaMA3.1-8B-Instruct model using the GSM8K and MATH validation sets, with results reported as normalized accuracy. As shown in Figure 10, increasing the SVD decomposition rank and employing higher-precision quantization consistently improve performance. Notably, the Attention module is more amenable to compression, achieving 100% performance retention with a rank of 1000 under double-precision settings. In contrast, the MLP module requires higher compression costs to reach comparable retention, highlighting the effectiveness of our proposed module-aware adaptive strategy. Quantization-related parameters, such as bit-width, are kept consistent with prior work, specifically Delta-CoMe (Ping et al., 2024), which employs training-free delta compression with mixed precision. Pruning ratios and the energy-preserving threshold $\epsilon$ are determined via simple grid search to balance compression efficiency and model performance.

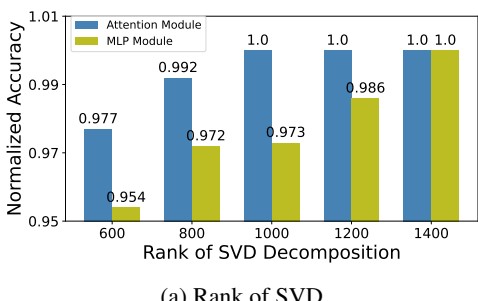
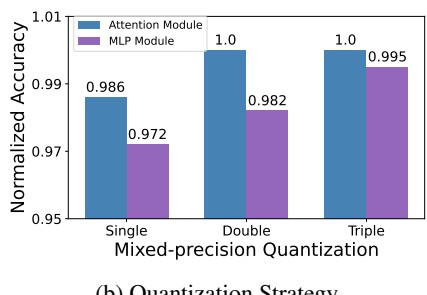

(a) Rank of SVD                                   (b) Quantization Strategy

Figure 10: Performance trends across hyperparameters using LLaMA3.1-8B-Instruct.

### B.3   Router for SkillPacks

The routing function $R$ is conditioned on either the source model or the task type and dynamically assigns each *SkillPack* to the target model. In the case of explicit knowledge fusion, the router is implemented as a lightweight feed-forward network consisting of four fully connected layers (input $\rightarrow 4096 \rightarrow 1024 \rightarrow 256 \rightarrow 5$ outputs), equipped with GELU activations and LayerNorm. The input features are drawn from the Llama2-7B embedding head, whose 4096-dimensional hidden representation is directly fed into the router. The router is trained using the same dataset employed for source-model distillation, with a batch size of 256, a learning rate of 0.001, and approximately 5,000 training steps. We train the router using the training datasets provided by FuseChat 2.0 (Wan et al., 2024b). Specifically, we collect the training loss on this dataset from five target models, each obtained through pairwise distillation from a different source model. These loss values serve as supervision signals, while input features are extracted from the embedded representations of the input data. A five-way classifier is then trained to predict the most suitable source model for each input. To evaluate the effect of routing quality, we vary the amount of training data used for the classifier to obtain models of different capabilities, and analyze their impact on fusion performance, as shown in Table 8. Since performance differences among source models are more pronounced on the MT-Bench dataset, the improvements from routing are more significant compared to AlpacaEval 2.0. For implicit knowledge fusion and forget-free learning experiments, we directly assign different *SkillPacks* based on task types, without training an additional router.

Table 8: Impact of the volume of training data on router effectiveness in explicit LLM fusion.

| Number of Samples | 5000 | 10000 | 20000 | 90000 |
|---|---|---|---|---|
| MT-Bench | 7.28 | 7.36 | 7.56 | **7.70** |
| AlpachaEval 2.0 (LC Win Rate) | 20.37 | 23.46 | 25.31 | **25.42** |

## C   Additional Results

### C.1   Result details of Knowledge Fusion an Forget-free Learning

We provide additional details to supplement our previous experiments, including the results of pairwise distillation and subsequent task vector compression in the heterogeneous knowledge fusion setting, as shown in Table 9 for explicit knowledge fusion and Table 10 for implicit knowledge fusion. These results demonstrate that our proposed `GraftLLM` framework effectively preserves performance in most cases.

In addition, we present more details on the forget-free learning experiments in Table 11, further highlighting the extent of catastrophic forgetting in forget-free learning scenarios and showcasing the advantages of our method. Finally, we include visualizations related to knowledge fusion in Figure 11 and Figure 12, offering a more intuitive understanding of the behavior of different methods.

### C.2   Detoxification with GraftLLM

We validate the effectiveness of our proposed `GraftLLM` method in the detoxification setting. Specifically, we extract a detachable *SkillPack* from the detoxified model obtained through knowledge

Table 9: Result details of explicit knowledge fusion on AlpacaEval 2.0 and MT-Bench. Reported metrics include pairwise distillation, pairwise distillation with modular-aware adaptive compression at 10% storage cost, and the performance retention ratio (%).

| Model | #Params | Pairwise Distillation | | | Pairwise Distillation + Compression | | |
|---|---|---|---|---|---|---|---|
| | | AlpacaEval 2.0 | | MT-Bench | AlpacaEval 2.0 | | MT-Bench |
| | | Win Rate | LC Win Rate | Average Score | Win Rate$_{(\%)}$ | LC Win Rate$_{(\%)}$ | Average Score$_{(\%)}$ |
| OpenChat-3.5-7B Starling | 7B | 11.28 | 16.14 | 7.22 | $11.26_{(99.8)}$ | $16.06_{(99.5)}$ | $7.20_{(99.7)}$ |
| OpenChat-3.5-7B SOLAR | 7B | 11.22 | 16.24 | 7.16 | $11.15_{(99.3)}$ | $16.21_{(99.8)}$ | $7.16_{(100)}$ |
| OpenChat-3.5-7B InternLM | 7B | 11.93 | 15.33 | 7.23 | $11.54_{(96.7)}$ | $15.33_{(100)}$ | $7.12_{(98.5)}$ |
| OpenChat-3.5-7B Mixtral | 7B | 11.71 | 16.46 | 7.28 | $11.41_{(97.4)}$ | $16.23_{(98.6)}$ | $7.24_{(99.5)}$ |
| OpenChat-3.5-7B Qwen | 7B | 11.13 | 15.10 | 7.23 | $11.13_{(100)}$ | $15.03_{(99.5)}$ | $7.17_{(99.2)}$ |

Table 10: Result details of implicit knowledge fusion on various tasks. Reported metrics include pairwise distillation, pairwise distillation with modular-aware adaptive compression at 8% storage cost, and the performance retention ratio (%). For tasks where performance degrades after distillation, we report results using the original target model instead.

| Model | Pairwise Distillation | | | Pairwise Distillation + Compression | | |
|---|---|---|---|---|---|---|
| **General Tasks** | | | | | | |
| | MMLU-Pro | MMLU-redux | GPQA-Dia | MMLU-Pro | MMLU-redux | GPQA-Dia |
| LLaMa-3.1-8B-Instruct | 51.3 | 73.0 | 37.7 | $51.3_{(100)}$ | $73.0_{(100)}$ | $37.7_{(100)}$ |
| Qwen-2.5-7B-Instruct | 55.6 | 76.6 | 38.1 | $55.4_{(99.6)}$ | $76.2_{(99.5)}$ | $38.1_{(100)}$ |
| **Mathematics Tasks** | | | | | | |
| | GSM8K | MATH | AMC23 | GSM8K | MATH | AMC23 |
| LLaMa-3.1-8B-Instruct | 88.8 | 56.2 | 37.5 | $88.2_{(99.3)}$ | $55.9_{(99.5)}$ | $35.0_{(93.3)}$ |
| Qwen-2.5-7B-Instruct | 92.6 | 75.3 | 57.5 | $92.0_{(99.4)}$ | $75.0_{(99.6)}$ | $55.0_{(95.7)}$ |
| **Code Tasks** | | | | | | |
| | HumanEval | MBPP | | HumanEval | MBPP | |
| LLaMa-3.1-8B-Instruct | 72.0 | 73.0 | | $72.0_{(100)}$ | $72.8_{(99.7)}$ | |
| Qwen-2.5-7B-Instruct | 85.7 | 84.8 | | $85.6_{(99.9)}$ | $84.5_{(99.6)}$ | |
| **Instruction Following** | | | | | | |
| | AlpacaEval 2.0 | MT-Bench | | AlpacaEval 2.0 | MT-Bench | |
| LLaMa-3.1-8B-Instruct | 65.4 | 9.0 | | $64.8_{(99.1)}$ | $8.8_{(97.8)}$ | |
| Qwen-2.5-7B-Instruct | 63.6 | 9.0 | | $61.5_{(96.6)}$ | $8.7_{(96.7)}$ | |

Figure 11: Comparison of explicit knowledge fusion methods for heterogeneous LLMs on AlpacaEval 2.0, including parameter size analysis.

Figure 12: A comparison of implicit knowledge fusion methods for heterogeneous LLMs with Qwen-2.5-7B-Instruct as target model.

editing. This modular SkillPack can be seamlessly integrated into the base model, enabling it to retain both detoxification capability and general-purpose performance. Our experiments are conducted on the mainstream chat model LLaMA3-8B-Instruct (Dubey et al., 2024b). We select three existing knowledge editing methods as baselines: **FT-L** (Meng et al., 2022), **WISE** (Wang et al., 2024e), and **DINM** (Wang et al., 2024d). Other common approaches, such as **ROME** (Mitchell et al., 2022) and **MEMIT** (Meng et al., 2023), require identifying specific model regions based on knowledge entities for parameter modification, making them less suitable for LLM detoxification tasks.

Table 11: Result details of continual learning on code and math tasks with modular-aware adaptive compression at 10% storage cost.

| Method | Additional Parameters | Original task (Code) | | | | New task (Math) | | | Average |
|---|---|---|---|---|---|---|---|---|---|
| | | HumanEval | HumanEval+ | MBPP | MBPP+ | GSM8K | MATH | AMC23 | |
| LLaMA3.1-8B-Instruct | - | 68.3 | 61.6 | 66.9 | 54.8 | 85.9 | 50.7 | 25.0 | 59.0 |
| Sequential Distillation | - | 69.1 | 63.2 | 67.4 | 55.9 | 87.5 | 55.7 | 30.0 | 61.3 |
| Distillation on Code | - | 72.0 | 65.2 | 73.0 | 62.7 | 85.2 | 50.3 | 22.5 | 61.6 |
| After Compression | 803M | 72.0 | 65.2 | 72.2 | 61.8 | 85.2 | 50.1 | 20 | 61.3 |
| Distillation on Math | - | 67.8 | 60.8 | 66.2 | 54.3 | 88.8 | 56.2 | 37.5 | 61.7 |
| After Compression | 803M | 67.4 | 60.2 | 66.2 | 54.1 | 88.2 | 55.9 | 35.0 | 61 |
| Multi LoRA r256 | 1.48B | 68.8 | 61.8 | 67.7 | 55.6 | 86.2 | 51.3 | 25.0 | 59.5 |
| Model Grafting[ICML23] | 803M | 70.4 | 63.9 | 69.1 | 57.5 | 87.2 | 53.4 | 27.5 | 61.3 |
| Model Tailor[ICML24] | 803M | 71.4 | 64.2 | 71.1 | 59.4 | 87.6 | 54.5 | 27.5 | 62.2 |
| **GraftLLM (ours)** | 803M | **72.0** | **65.2** | **72.2** | **61.8** | **88.2** | **55.9** | **35.0** | **64.3** |

We conduct our experiments on the `SafeEdit` benchmark (Wang et al., 2024d) using the `EasyEdit` framework (Wang et al., 2023). For all methods involving training components, we utilize the training and validation sets for model development and evaluate the final performance on the test set. As shown in Table 13, our method achieves detoxification performance comparable to DINM, while better preserving the general capabilities of the base model. Overall, `GraftLLM` outperforms the previous best approach by *1.76* points in terms of the average score across detoxification and general tasks.

## C.3 CASE STUDIES ON SKILLPACK KNOWLEDGE

We present qualitative examples illustrating how the router handles inputs that combine biomedical, financial, and legal terminology, as shown in Tab. 12. For each input, we report the selected SkillPack and the corresponding domain knowledge it encodes. As shown in the table below, the router consistently selects the SkillPack aligned with the dominant semantic cue, and each SkillPack captures coherent, domain-specific knowledge—for instance, dose–response reasoning in Biomedicine and regulatory interpretation in Law. These examples provide a clear qualitative understanding of the information encoded in each SkillPack and how the router effectively manages mixed-domain queries.

Table 12: Qualitative examples illustrating router behavior and the knowledge captured by SkillPacks.

| Input Example | Router (Top-1) | Knowledge Captured in SkillPack |
|---|---|---|
| Does increasing the dose improve overall return under the current treatment policy? | Biomed | Captures clinical reasoning such as dose–response relationships, treatment efficacy, and pharmacological effects. |
| What regulatory constraints apply when reporting adverse outcomes that may affect financial liability? | Law | Encodes legal concepts including regulatory compliance, reporting requirements, and liability interpretation. |
| Evaluate whether the compound shows significant impact on long-term yield. | Biomed | Stores biochemical and pharmacological knowledge regarding compound effects, biological impact, and experimental outcomes. |

## D BASELINES DETAILS

This section provides a detailed description of the baseline, as outlined below.

- **FuseLLM** (Wan et al., 2024a) is the first to introduce multi-teacher distillation for fusing knowledge from heterogeneous large language models.
- **FuseChat 2.0** (Wan et al., 2024b) fuses chat LLMs of different scales and structures through lightweight pairwise fine-tuning into target models of the same size. It uses a statistics-based token alignment for compatibility and merges the targets in parameter space.

Table 13: Detoxification performance and general performance of vanilla LLMs and various detoxification methods on SafeEdit. The detoxification performance (detoxification success rate) is multiplied by 100. DG-Avg represents the average performance across the four DG metrics. The **best** and underline{second-best} results on each model are highlighted in **bold** and underlined, respectively.

| Method | Detoxification Performance (↑) | | | | | | General Performance (↑) | | | Average (↑) |
|---|---|---|---|---|---|---|---|---|---|---|
| | DS | $DG_{onlyQ}$ | $DG_{otherA}$ | $DG_{otherQ}$ | $DG_{otherAQ}$ | DG-Avg | Fluency | KQA | CSum | |
| LLaMA3-8B-Instruct | 14.82 | 55.41 | 31.14 | 13.88 | 31.43 | 32.97 | **7.89** | **64.83** | **25.81** | 29.26 |
| FT-L[NeurIPS22] | 82.18 | 97.75 | 90.90 | 79.83 | 93.81 | 90.57 | 6.42 | 63.03 | 25.51 | 53.54 |
| WISE[NeurIPS24] | 81.43 | 81.24 | 81.99 | 68.86 | 80.30 | 78.10 | 5.64 | 62.99 | 25.90 | 50.81 |
| DINM[ACL24] | **82.89** | **99.24** | **98.87** | **99.70** | **99.78** | **99.40** | 1.20 | 62.98 | 25.18 | 54.33 |
| **GraftLLM (ours)** | 82.83 | 98.84 | 98.46 | **99.70** | 99.34 | 99.08 | 7.89 | 64.83 | 25.81 | **56.09** |

- **FuseChat 3.0** (Yang et al., 2025b) further introduces implicit model fusion and a DPO-based strategy to enhance alignment and integration performance across heterogeneous LLMs.

- **Task Arithmetic** (Ilharco et al., 2023) first defines the concept of "task vectors" and merges these vectors into a pre-trained model to execute multi-task learning. The model is produced by scaling and adding the task vectors to the initial model as $\theta_m = \theta_{init} + \lambda * \sum_{t=1}^{n} \tau_t$.

- **Ties-Merging** (Yadav et al., 2024) further solves the task conflict problem in Task Arithmetic (Ilharco et al., 2023). It eliminates redundant parameters and resolves symbol conflicts through three steps: Trim, Elect Sign, and Disjoint Merge.

- **DARE** (Yu et al., 2023) sets the majority of delta parameters to zero and rescale the rest by $\theta' = \theta \cdot (1/(1-p))$ where $p$ is the proportion of delta parameters dropped, therefore efficiently reduces parameter redundancy.

- **LoraHub** (Huang et al., 2023) employs Low-rank Adaptations to dynamically combine task-specific modules for cross-task generalization, and adapts to new tasks by configuring $\theta' = \sum_{k=1}^{K} w_k \cdot \theta_k$.

- **PCB-Merging** (Du et al., 2024b) effectively adjusts parameter coefficients through balancing parameter competition within model population.

- **InfiFusion** (Yan et al., 2025) enhances Universal Logit Distillation with Top-K selection and Logits Standardization to improve cross-model alignment. Top-K filters noisy outputs, while standardization ensures consistent logit distributions across diverse models.

- **TALL-Mask** (Wang et al., 2024c) localize the task-specific information in a multi-task vector, which deactivates irrelevant parts for each task in the merged multi-task vector with binary masks.

- **EMR-Merging** (Huang et al., 2024) first selects a unified model from all weights, then generates lightweight task-specific modulators—masks and rescalers—to align direction and magnitude with each source model.

- **Delta-CoMe** (Ping et al., 2024) propose a mixed-precision delta-compression method that employs varying bit-widths for different singular vectors based on their singular values

- **Model Grafting** (Panigrahi et al., 2023) introduces the concept of skill localization—identifying where task-specific skills reside within the model—and proposes a method to efficiently acquire them.

- **Model Tailor** (Zhu et al., 2024) derives a sparse mask to identify the "model patch" through a fusion of salience and sensitivity analysis, and then decorates the patch to enhance performance.

# E DATASETS DETAILS

## E.1 TRAING DATASETS FOR EXPLICIT KNOWLEDGE FUSION

We use a comprehensive training dataset, FUSECHAT-MIXTURE (Wan et al., 2024b), from various sources. This dataset covers different styles and capabilities, featuring both human-written and

model-generated, and spanning general instruction-following and specific skills. These sources include:

**Orca-Best**[1]: We sampled 20,000 examples from Orca-Best, which is filtered from the GPT-4 (1M) partition of Orca (Mukherjee et al., 2023) based on maximum length and clustering of instructions.

**Capybara**[2]: We incorporated all the 16,000 examples of Capybara, which is a high-quality collection of multi-turn synthetic conversations.

**No-Robots**[3]: We included all the 9,500 examples of No-Robots, which is a dataset created by skilled human annotators for supervised fine-tuning.

**ShareGPT-GPT4**[4]: We utilized all 6,200 examples from ShareGPT-GPT4, which exclusively uses dialogues generated by GPT-4 in ShareGPT.

**Oasst-Top1**[5]: We selected 5,000 examples from Oasst-Top1, which is a refined version of Oasst1 (Köpf et al., 2024), a human-annotated assistant-style conversation dataset.

**MetaMathQA** [6]: We sampled 10,000 examples from MetaMathQA (Yu et al., 2024), which is augmented from the GSM8K (Li et al., 2025) and MATH (Hendrycks et al., 2021b) datasets for mathematics problem-solving.

**OSS-Instruct** [7]: We chose 10,000 examples from OSS-Instruct (Hao et al., 2026), which contains code instruction data synthesized from open-source code snippets.

**Evol-Alpaca** [8]: We sampled 10,000 examples from Evol-Alpaca, which is a code instruction dataset generated by GPT-4 with evol-instruct proposed by WizardCoder (Luo et al., 2024).

**Python-Code** [9]: We selected 10,000 examples from Python-Code, which comprises instructions and responses generated by GPT-3.5 and GPT-4 for python code generation.

We followed the data processing code in FastChat (Zheng et al., 2024a) to clean instances containing non-English or special characters. Then, we split long conversations into blocks with a maximum length of 2048 tokens, resulting in the final FUSECHAT-MIXTURE with 95,000 samples.

## E.2   TRAING DATASETS FOR IMPLICIT KNOWLEDGE FUSION

The training datasets used in the implicit knowledge experiments are listed in Table 14. Additionally, we provide the Hugging Face repository names and corresponding links for the target LLMs, source LLMs, and the reward model.

## E.3   EVALUATION BENCHMARKS

**AlpacaEval-2** (Li et al., 2023b) comprises 805 instructions from five different datasets and assesses models using two metrics: length-controlled (LC) win rate and raw win rate (WR) (Dubois et al., 2024). GPT-4-Preview-1106 serves as both the baseline model and the evaluator for the other models.

**MT-Bench** (Zheng et al., 2023) contains 80 multi-turn dialogues across eight categories, including writing, roleplay, reasoning, math, coding, extraction, STEM, and humanities. Each response is evaluated by GPT-4 on a scale from 1 to 10, with the average score reported for each dialogue turn across the 80 dialogues. We use GPT-4-0613 as the judge model following the official setting.

**MMLU-Pro** (Wang et al., 2024g) is an enhanced version of the MMLU dataset, designed to address issues such as noisy data and reduced difficulty due to advances in model capabilities and

---

[1] https://huggingface.co/datasets/shahules786/orca-best
[2] https://huggingface.co/datasets/LDJnr/Capybara
[3] https://huggingface.co/datasets/HuggingFaceH4/no_robots
[4] https://huggingface.co/datasets/shibing624/sharegpt_gpt4
[5] https://huggingface.co/datasets/OpenAssistant/oasst_top1_2023-08-25
[6] https://huggingface.co/datasets/meta-math/MetaMathQA
[7] https://huggingface.co/datasets/ise-uiuc/Magicoder-OSS-Instruct-75K
[8] https://huggingface.co/datasets/theblackcat102/evol-codealpaca-v1
[9] https://huggingface.co/datasets/ajibawa-2023/Python-Code-23k-ShareGPT

Table 14: Details of open-source models and datasets used in Implicit Knowledge Fusion.

| Name | Huggingface ID |
|------|----------------|
| **Target LLMs** | |
| Llama-3.1-8B-Instruct | meta-llama/Llama-3.1-8B-Instruct |
| Qwen-2.5-7B-Instruct | Qwen/Qwen2.5-7B-Instruct |
| **Source LLMs** | |
| Mistral-Large-Instruct-2407 | Mistral-Large-Instruct-2407 |
| Gemma-2-27B-it | google/gemma-2-27b-it |
| Qwen-2.5-72B-Instruct | Qwen/Qwen2.5-72B-Instruct |
| Llama-3.1-70B-Instruct | meta-llama/Llama-3.1-70B-Instruct |
| **Reward Model** | |
| ArmoRM-LLaMA3-8B-v0.1 | RLHFlow/ArmoRM-Llama3-8B-v0.1 |
| **Datasets** | |
| UltraFeedback | princeton-nlp/llama3-ultrafeedback-armorm |
| Magpie-Pro-DPO | Magpie-Align/Magpie-Llama-3.1-Pro-DPO-100K-v0.1 |
| HelpSteer2 | nvidia/HelpSteer2 |
| OpenMathInstruct-2 | nvidia/OpenMathInstruct-2 |
| LeetCode | greengerong/leetcode |
| Self-Oss-Instruct-SC2 | bigcode/self-oss-instruct-sc2-exec-filter-50k |
| Alpaca-GPT4-Zh | llamafactory/alpaca_gpt4_zh |
| Magpie-Qwen2-Pro-Zh | Magpie-Align/Magpie-Qwen2-Pro-200K-Chinese |

increased data contamination. MMLU-Pro increases challenge levels by expanding multiple-choice options from 4 to 10, requiring reasoning across more questions, and incorporating expert-reviewed annotations for improved quality and reduced noise.

**MMLU-redux** (Gema et al., 2024) is a re-annotated subset of the MMLU dataset created through manual assessment from 14 human experts. **GPQA-Diamond** (Rein et al., 2023) is a challenging knowledge benchmark crafted by PhD-level domain experts in biology, physics, and chemistry. The dataset contains questions that are straightforward for experts but difficult for laypersons. We evaluate the highest quality diamond set comprising 198 questions.

**Arena-Hard** (Li et al., 2024b) is a challenging instruction-following benchmark that closely aligns with the human preference ranking from Chatbot Arena, a crowd-sourced platform for evaluating LLMs. It spans 250 high-quality topic clusters including 500 well-defined technical problem-solving queries. We report the win rate against GPT-4-0314 using GPT-4-Preview-1106 as the judge model. **GSM8K** (Cobbe et al., 2021) is a set of grade-school math word questions that evaluates mathematical reasoning capabilities.

**MATH** (Hendrycks et al., 2021a) is a dataset of math problems ranging in difficulty from middle school to high school competition level. It tests a wide range of mathematical skills, including algebra, calculus, number theory, and probability.

**AMC 23**[10] (Yang et al., 2024b) refers to the 2023 American Mathematics Competition, featuring 25 multiple-choice questions that test advanced high school mathematics, including trigonometry, advanced algebra, and elements of calculus.

**HumanEval** (Chen et al., 2021) evaluates code generation capabilities by presenting models with function signatures and docstrings and requiring them to implement the function body in Python.

**MBPP** (Austin et al., 2021) is a dataset of simple programming problems designed to assess the ability of models to generate short Python code snippets from natural language descriptions.

---

[10]https://huggingface.co/datasets/AI-MO/aimo-validation-amc

# F   IMPLEMENTATION DETAILS

## F.1   TRAINING DETAILS

**Explicit Knowledge Fusion**    In this experiment, we primarily focus on effectively integrating chat LLMs with diverse architectures and varying model sizes. We select six representative source models: `OpenChat-3.5-7B` (Wang et al., 2024a), `Starling-LM-7B-alpha` (Zhu et al., 2023), `NH2-SOLAR-10.7B` (Kim et al., 2023), `InternLM2-Chat-20B` (Cai et al., 2024), `Mixtral-8x7B-Instruct` (Wang et al., 2024a), and `Qwen-1.5-Chat-72B` (Bai et al., 2023). We use `OpenChat-3.5-7B` as the pivot model and starting point for generating target LLMs, given its well-balanced size and performance. Initially, we apply pairwise knowledge fusion to produce five target models with uniform architecture. Subsequently, these target models' knowledge is combined through either parameter merging or routing mechanisms.

**Implicit Knowledge Fusion**    The construction of data plays a vital role in facilitating the Implicit Model Fusion approach showcased in FuseChat-3.0 (Yang et al., 2025b). We perform SFT+DPO on four task-specific datasets to obtain **four individually fine-tuned models**, each achieving strong performance on its corresponding task. We describe the procedures for selecting prompts, sampling responses, and assembling the dataset, explaining the reasoning behind each design choice.

- **Prompt Selection**: To enhance the target LLMs' abilities across multiple fields—including instruction following, math, coding, and Chinese—we assemble a diverse dataset. This is done by carefully choosing samples from well-regarded open-source community datasets, followed by specific filtering and preprocessing steps to ensure quality and relevance.

- **Response Sampling**: For each prompt in the curated dataset, we generate responses primarily from four leading source LLMs. Our response sampling strategy is tailored to each domain, leveraging `vLLM`[11] (Zhu et al., 2023) as the inference backend. We perform multiple sampling runs using different random seeds to ensure diversity. The sampling parameters for each source model are as follows: for `Gemma-2-27B-it`, `Mistral-Large-Instruct-2407`, and `Llama-3.1-70B-Instruct`, we set the temperature to 0.8 and top-p to 0.95; for `Qwen-2.5-(Math)-72B-Instruct`, we use a temperature of 0.7, top-p of 0.8, and a repetition penalty of 1.05.

- **Preference Pairs**: To construct preference pairs from models with diverse output styles, we select the best and worst responses generated by the same source model for each pair. This intra-model pairing strategy reduces reward bias caused by heterogeneous response styles, prevents reward hacking, and provides a more controlled and reliable preference signal. The data construction process varies by domain: for instruction-following and conversational data, we use an external reward model to evaluate responses; for mathematics and coding domains, responses are verified through rule-based systems.

The final dataset $\mathcal{D}$ consists of 158,667 entries, with 94,539 allocated to the SFT phase ($\mathcal{D}_{\text{SFT}}$) and 64,128 preference pairs for the DPO phase ($\mathcal{D}_{\text{DPO}}$). The dataset composition is provided in Table 15.

## F.2   HYPERPARAMETER SETTINGS

**Explicit Knowledge Fusion**    we follow a same experiment setting as FuseChat 2.0 (Wan et al., 2024b), we train the target LLMs using a batch size of 128 and a maximum length of 2048 on a single node with 8x80GB NVIDIA A800 GPUs for three epochs, which takes approximately 9 hours. The models are optimized using the AdamW (Loshchilov & Hutter, 2019) optimizer with $\beta_1 = 0.9$ and $\beta_2 = 0.999$. We use a weight decay of 0.0 and gradient clipping of 1.0. A cosine learning rate schedule is employed, with a maximum learning rate of 5e-6 and a warmup ratio of 0.03.Our training framework is implemented based on the HuggingFace Transformers (Wolf et al., 2020).

**Implicit Knowledge Fusion**    In our SFT experiments, we use the Llama-Factory library[12] (Zheng et al., 2024b) to implement the fine-tuning. For all target models, we perform fine-tuning for 3 epochs, with a batch size of 128 and a maximum sequence length of 2048 tokens. A cosine learning rate

---

[11]https://github.com/vllm-project/vllm
[12]https://github.com/hiyouga/LLaMA-Factory

Table 15: The constitution of Implicit Knowledge Fusion dataset in SFT phase and DPO phase. As no suitable reward models were available for Chinese, we used all samples for SFT and omitted the DPO phase.

| Category | Dataset | Count | $\#\mathcal{D}_{\text{SFT}}$ | $\#\mathcal{D}_{\text{DPO}}$ |
|---|---|---|---|---|
| Instruction Following | UltraFeedback | 51,098 | 20,439 | 30,659 |
| | Magpie-Pro-DPO | 20,374 | 8,149 | 12,225 |
| | HelpSteer2 | 9,435 | 3,774 | 5,661 |
| Mathematics | OpenMathInstruct-2 | 51,803 | 40,188 | 11,615 |
| Coding | LeetCode | 3,113 | 1,877 | 1,236 |
| | Self-Oss-Instruct-SC2 | 12,892 | 10,160 | 2,732 |
| Chinese Language | Alpaca-GPT4-Zh | 2,471 | 2,471 | 0 |
| | Magpie-Qwen2-Pro-Zh | 7,481 | 7,481 | 0 |
| *Total* | | 158,667 | 94,539 | 64,128 |

Table 16: Hyperparameters for different target models during the SFT and DPO stages.

| Target Model | SFT Learning Rate | DPO Learning Rate | DPO $\lambda$ | DPO Loss Type |
|---|---|---|---|---|
| Llama-3.1-8B-Instruct | $5 \times 10^{-6}$ | $8 \times 10^{-7}$ | 10 | $\mathcal{L}_{\text{LN-DPO}}$ |
| Qwen-2.5-7B-Instruct | $2 \times 10^{-6}$ | $3 \times 10^{-7}$ | 0.01 | $\mathcal{L}_{\text{DPO}}$ |

schedule with a warmup ratio of 0.1 is employed. In DPO experiments, we utilize the alignment-handbook[13] as the training framework for DPO. All post-SFT target models undergo training for one epoch with a batch size of 128 and a maximum sequence length of 2048. A cosine learning rate schedule with a warmup ratio of 0.1 is used. Checkpoints are saved every 100 steps, and the best checkpoint from the last two is selected. The hyperparameter configurations for different models are detailed in Table 16.

### F.3    COMPUTATIONAL RESOURCES AND RUNTIMES

We report the resource consumption and runtime of our module-adaptive compression strategy in Table 10. The overall compression overhead is calculated as the weighted sum of each module's storage cost, with weights corresponding to the proportion of parameters in each module. Additionally, we provide the compression time for each module, which mainly depends on the parameter count and the amount of data used for GPTQ (Frantar et al., 2022). In all our experiments, we use the C4 validation split [14] as the calibration set for GPTQ, which is widely adopted in previous GPTQ-based quantization research. Since pruning and SVD typically require only a few minutes, we omit their time costs and primarily report the runtime for quantization. In practice, the compression strategy can be adjusted according to task complexity and model type.

Table 17: Storage and time costs of our proposed adaptive compression strategy. The Storage Cost is defined as the ratio of the compressed module size to that of the corresponding original parameters.

| Modules | Compression Strategy | Quantization Strategy | Storage Cost (%) | Time Cost |
|---|---|---|---|---|
| Embedding & Head | Pruning with $\alpha = 0.5$ | 4bits | 12.5 | 8 minutes |
| MLP Module | SVD with $r = 1400$ | [8, 3, 2] bits for rank [20, 180, 1200] | 5.43 | 53 minutes |
| Attention Module | SVD with $r = 1000$ | [8, 2] bits for rank [20, 980] | 5.59 | 22 minutes |

In addition, we provide a detailed comparison of training and inference resource usage across different methods (see Tab. 18). In all experiments, we adopt top-1 router activation, so that during inference, the computational overhead primarily comes from the router forward pass and the SkillPack decompression step. Both of these steps introduce a negligible increase in latency, accounting for about

---

[13]https://github.com/huggingface/alignment-handbook
[14]https://huggingface.co/datasets/allenai/c4/en/c4-validation.

3% of the total inference time, demonstrating that our method maintains efficiency comparable to baseline models.

Table 18: Resource analysis of GraftLLM compared to representative LLM fusion baselines.

| Method | Model Size | Router Size | Training Cost (GPU h) | Compression Cost | Inference Overhead |
|---|---|---|---|---|---|
| FuseLLM | 7B | – | 132 h | – | 1.00× |
| FuseChat | 7B | – | 132 h | 4 min | 1.00× |
| Twin-Merging | 21B | 21M | 132 h + 8 min | 9 min | 1.03× |
| **GraftLLM (Ours)** | 9.2B | 21M | 132 h + 8 min | 163 min | 1.03× |

## G  THE USE OF LARGE LANGUAGE MODELS (LLMS)

During the preparation of this manuscript, large language models were used solely for minor stylistic enhancements and occasional grammatical corrections. All conceptual insights, analytical reasoning, and interpretive conclusions were generated entirely by the authors. No algorithmic assistance was sought in the formulation, design, or substantive content of the work, and full scientific responsibility lies solely with the human contributors.

