# OpenReview forum: "Knowledge Fusion of Large Language Models via Modular SkillPacks"
_ICLR.cc/2026/Conference — ICLR 2026 Poster_

### Official Review · Reviewer_ZGH4 · 2025-10-21

**Soundness:** 3
**Presentation:** 2
**Contribution:** 3
**Rating:** 6
**Confidence:** 4

**Summary:**

This paper introduces GraftLLM for transferring capabilities from heterogeneous source LLMs into a single target LLM. The key contribution of GraftLLM is to compress task-specific knowledge distilled from various source models into compact, module-wise "SkillPacks", effectively preserving the target model's knowledge and preventing catastrophic forgetting. These lightweight SkillPacks can be flexibly routed and integrated into the target model, offering a plug-and-play method for practical deployment. Comprehensive experiments span pairwise grafting, knowledge fusion and forget-free learning demonstrate the effectiveness of GraftLLM.

**Strengths:**

1. This paper introduces an innovative framework that creatively combines knowledge fusion, model compression, and model rounting. The introduced "SkillPack" is flexible and plug-and-play, offering a cost-effective solution for scalable, efficient, and forget-free knowledge transfer across heterogeneous LLMs.
2. The work is technically strong. The proposed module-aware adaptive compression strategy is well-motivated and well-designed to preserve performance while minimizing parameter overhead. The experimental methodology is rigorous, featuring comprehensive comparisons against various baselines across three distinct scenarios.

**Weaknesses:**

1. The presentation of the paper has several issues that hinder readability. These include, but are not limited to:

- **Grammatical errors**: For example, the word "heterogeneous" is repeated on line 26.

- **Mismatched citations**: For instance, the citations for FuseChat  (Yang et al., 2025b) on line 54 and synthetic data  (Wan et al., 2024b) on line 72 appear to be swapped.

- **Mismatched table captions**: The caption for Table 6 does not correspond to its content. The caption of Table 10 should appear before the table.

- **Logical contradiction**: Appendix B.3 states, "A five-way classifier is then trained to predict the most suitable source model for each input," which suggests the routing function $\mathcal{R}$ selects the best SkillPack on a per-prompt basis. However, this appears to conflict with the description in Section 3.3 (lines 270-272), which defines $\mathcal{R}$ as a submodule-specific router that "dynamically assigns each SkillPack to the appropriate submodules." The authors should rewrite Section 3.3 to clarify this mechanism and resolve the ambiguity.

2. Several experimental details are missing, which poses a challenge for reproducibility. These omissions include:

- **Insufficient details about the router**: For the explicit fusion experiments, the paper mentions training a router to assign SkillPacks. However, the details of this router (e.g., its architecture, the distribution of training data, and its computational overhead) are too brief.

- **Lack of detail on model combination**: Section 3.2 explains how a SkillPack is created through compression, but the paper does not explicitly describe how these compressed delta parameters are combined with the full parameters of the target model to form the final, fused model.

- **Missing details for pairwise distillation in implicit fusion**: Section 5.2 describes experiments on implicit fusion, which involves a pairwise distillation step. However, the implementation details provided in Appendix F.1 (lines 1269-1318) appear to describe the process for the FuseChat-3 baseline, not the specific distillation process used to generate the SkillPacks for GraftLLM in this setting.

- **Insufficient detail on hyperparameter selection**: The module-aware adaptive compression strategy described in Section 3.2 introduces several tunable hyperparameters (e.g., SVD rank, pruning ratio, quantization bitwidth, energy threshold $\beta$). However, the paper does not provide enough detail on how the final values for these hyperparameters were selected.

**Questions:**

1. What is the inference overhead of the proposed method? Specifically, does loading multiple SkillPacks significantly increase inference latency? Additionally, what is the computational cost associated with training the router?

2. For training the router module in explicit model fusion, what is the rationale for using training loss values as supervision signals? Furthermore, how is the accuracy of this routing mechanism ensured, and have you analyzed whether a potential distribution shift between the router's training data and the test set could affect its performance?

3. In the implicit fusion experiments (Table 2), could you clarify how the models for GraftLLM's pairwise distillation step were trained? What were the individual performances after distilling from each source model, before the final fusion? The overall performance gain over the FuseChat-3 baseline is modest; could you elaborate on the trade-off between this performance gain and the complexity/overhead of the GraftLLM approach in this specific scenario?

4. I am willing to raise my score if the authors can satisfactorily address the weaknesses and questions I have raised.

---

> ### Author Response · Authors · 2025-11-25
> **Response to Reviewer ZGH4 (part 1/4)**
>
> Thank you for your valuable comments. We will explain your concerns point by point.
>
> ---
>
> > **Weakness 1: Presentation and Writing Issues?**
>
> **Reply**: We sincerely thank the reviewer for pointing out these clarity and formatting issues. We have carefully addressed each of them in the revised manuscript. All grammatical mistakes and mismatched citations have been corrected, and the table captions have been aligned with their corresponding content. We also acknowledge the ambiguity between the router description in Appendix B.3 and Section 3.3. To resolve this, we have substantially rewritten Section 3.3 to clearly explain the two routing strategies, their purposes, and how they are used in different experimental settings. We believe these revisions significantly improve readability and eliminate the confusion highlighted by the reviewer.
>
> ---
>
> > **Weakness 2：Missing Experimental & Implementation Details.**
>
> **Reply**: **(2.1) Details regarding the router.** We have now fully clarified these components in the revised version. In the implicit model fusion setting, the router is implemented as a **lightweight feed-forward network** consisting of **four fully connected layers** (input → 4096 → 1024 → 256 → 5), with **GELU activations** and **LayerNorm**. The **4096-dim input** is taken directly from the **Llama2-7B embedding head**, whose hidden representation serves as the router’s input features. The router is trained using the **same dataset** as the one used for the pairwise distillation of source models. In the revision, we have added the previously missing **training hyperparameters**, including batch size, learning rate, and training steps. As stated in the original submission, the router receives supervision from the **per-input losses** of the five pairwise-distilled models, allowing it to predict the most suitable source model for each example. Training the router takes **approximately 8 minutes** on a single GPU. During inference, we adopt **top-1 routing**, which ensures that the inference-time computational cost is effectively the same as baseline methods, aside from the negligible router forward pass.
>
> **(2.2) SkillPack Composition Procedure.** In the revised version, we have explicitly added the full SkillPack composition procedure. After compression, each SkillPack is first **decoded through dequantization**, followed by **SVD reconstruction** to recover the corresponding task vector. The reconstructed delta parameters are then **added back to the full parameters of the base model**, producing the final fused model. This step was previously implicit, and we have now updated Section 3.2 to clearly describe this end-to-end process.
>
> **(2.3)** **Distillation Procedure In the implicit fusion**. We thank the reviewer for highlighting this missing detail. In the implicit fusion experiments, we first perform **SFT+DPO** on four task-specific datasets to obtain **four individually fine-tuned models**, each achieving strong performance on its corresponding task. In the original version, **Appendix C.1 and Table 10 reported the individual distilled performances** for transparency. The key difference between our approach and the baseline method lies in **decoupling and compressing task-specific capabilities**. This design allows us to construct a **lightweight, high-performance dynamic fusion framework** with minimal additional parameter overhead, unlike the baseline, which merges models without explicit capability disentanglement.
>
> **(2.4) Compression Hyperparameter Selection Details:** We appreciate the reviewer’s comment regarding hyperparameter selection. For the module-aware adaptive compression strategy, the quantization-related parameters (e.g., bit-width) are set consistently with prior work, specifically **Delta-CoMe (NeurIPS 2024)**, which uses training-free delta compression with mixed precision, while the pruning ratios and energy-preserving threshold ε are determined via a simple grid search to balance compression and performance. These details were already included in the appendix of the original submission, and we have now emphasized them in the revised version to improve clarity and reproducibility.

---

> ### Author Response · Authors · 2025-11-25
> **Response to Reviewer ZGH4 (part 2/4)**
>
> > **Question 1: Inference Overhead of SkillPack.**
>
> **Reply**: We thank the reviewer for raising this question. In the revised manuscript, we provide a detailed comparison of **training and inference resource usage** across different methods (see the table below). In all experiments, we adopt **top-1 router activation**, so that during inference, the computational overhead primarily comes from the **router forward pass** and the **SkillPack de-compression step**. Both of these steps introduce a negligible increase in latency, accounting for **about 3% of the total inference time**, demonstrating that our method maintains efficiency comparable to baseline models.
>
> Table 1: Training and inference resource comparison between SkillPack and representative LLM fusion baselines.
>
> |                         Method                         | Model Size | Router Size | Training Cost (GPU h) | Compression Cost | Inference Overhead |
> | :----------------------------------------------------: | ---------- | :---------: | :-------------------: | :--------------: | :----------------: |
> |          Multi-Teacher Distillation (FuseLLM)          | 7B         |      –      |         132 h         |        –         |       1.00×        |
> |  Pairwise Distillation + Parameter Merging (FuseChat)  | 7B         |      –      |         132 h         |      4 min       |       1.00×        |
> | Pairwise Distillation + Dynamic Merging (Twin-Merging) | 21B        |     21M     |     132 h + 8 min     |      9 min       |       1.03×        |
> |      **Pairwise Distillation + SkillPack (Ours)**      | 9.2B       |     21M     |     132 h + 8 min     |     163 min      |       1.03×        |

---

> ### Author Response · Authors · 2025-11-25
> **Response to Reviewer ZGH4 (part 3/4)**
>
> > **Question 2: Analysis of router behavior.**
>
> **Reply:** Using per-sample training losses as supervision for the router in explicit model fusion is intentional and reasonable. This signal allows the router to learn **which source model specializes in which type of input**, without relying on test-set labels or test-set performance, thus avoiding unfair evaluation leakage.
>
> Router behavior varies with task similarity, as shown in Table 2:
>
> - **Similar tasks (e.g., Explicit/Implicit LLM Fusion):** When SkillPacks represent highly similar capabilities, their decision boundaries overlap, making top-1 routing **intrinsically ambiguous**. Routing to multiple SkillPacks (e.g., top-2) can yield noticeable performance gains. However, each SkillPack must first be decompressed and its delta applied to **reconstruct full model parameters**, meaning the inference cost **scales** almost linearly with the number of SkillPacks. For efficiency and fairness, our main experiments use **top-1 routing**, activating only the most relevant SkillPack while **minimizing overhead** and retaining the benefits of modular, task-specific deltas.
> - **Divergent tasks (e.g., Finance–Bio–Law LLM Fusion):** When SkillPacks are strongly differentiated, the router easily maps inputs to the correct domain expert. Here, top-1 routing suffices, and ensembling provides little benefit because the knowledge is not mutually reinforcing.
>
> Regarding the concern about distribution shift: we conducted an additional analysis by training the router directly on the test set (i.e., using test-set losses as supervision). As expected, **accuracy improves**, but the computational cost becomes significantly higher because it requires evaluating all SkillPacks **on the test inputs** before router training, which is **impractical for deployment**. This further justifies our design choice of using training-set losses as supervision.
>
> Finally, in domains with large task gaps—such as our financial–legal–medical fusion experiment—the router trained on training-set losses already achieves highly **reliable selection**, demonstrating that this approach is effective and robust in practice.
>
> Table 2: Analysis of router behavior and multi-SkillPack ensembling across three fusion settings.
>
> |          Methods          | Explicit LLM Fusion (*AlpacaEval 2.0*) | Implicit LLM Fusion (*10 benchmark*) | Fusion Finance+Law+Bio LLM | Inference Time |
> | :-----------------------: | :------------------------------------: | :----------------------------------: | :------------------------: | -------------- |
> |         FuseChat          |             11.63 / 16.89              |                 55.2                 |           47.33            | 1x             |
> | Grafting Top 1 SkillPack  |  16.56 **(+4.9)** / 25.42 **(+8.5)**   |           56.0 **(+0.8)**            |     49.62 **(+2.29)**      | 1.03 x         |
> | Grafting Top 2 SkillPacks |      17.33 (+0.8) / 26.28 (+0.9)       |             56.3 (+0.3)              |     **49.64** (+0.02)      | 2.03 x         |
> | Grafting Top 3 SkillPacks |  **17.49** (+0.2) / **26.72** (+0.4)   |           **56.4** (+0.1)            |       49.63 (-0.01)        | 3.03 x         |

---

> ### Author Response · Authors · 2025-11-25
> **Response to Reviewer ZGH4 (part 4/4)**
>
> > **Question 3：Results on Implicit LLM fusion.**
>
> **Reply**: We appreciate the reviewer’s question. In the revised version, we have added detailed descriptions of how the four task-specific models used in the implicit fusion setting were trained, including the SFT+DPO procedure and the individual distilled performances before fusion. Regarding the modest improvement over the FuseChat-3 baseline: this particular setting is not an ideal scenario for evaluating GraftLLM. One of the expert models—trained on instruction-following data—exhibited very **strong general capabilities**, outperforming the other task-specific models on out-of-domain tasks. As a result, the benefits from incorporating the **math and code experts became less pronounced**, which naturally limited the performance gains. To better illustrate GraftLLM’s advantages, we include an additional experiment involving three more distinct domains (finance, law, and medicine). As shown in the new results Table 3, our method achieves significantly larger improvements and maintains **almost lossless performance** after fusion, demonstrating its effectiveness in settings with genuinely complementary expert capabilities.
>
> Table 3: Performance comparison between GraftLLM and representative LLM fusion baselines, evaluated across three domains: Biomedicine, Finance, and Law.
>
> |             Methods              | Params   | Biomedicine | Finance  |   Law    | Average        |
> | :------------------------------: | -------- | :---------: | :------: | :------: | -------------- |
> |             LLaMA-7B             | 7B       |    44.2     |   58.6   |   34.2   | 45.7           |
> |    AdaptLLM-7B [1]\[ICLR24\]     | **3*7B** |    47.3     |   63.4   |   38.5   | 49.7_(100%)    |
> |           LLaMA-7B-Bio           | 7.7B     |  **47.3**   |   57.9   |   34.5   | 46.6           |
> |         LLaMA-7B-Finace          | 7.7B     |    43.7     | **63.4** |   34.0   | 57.0           |
> |           LLaMA-7B-Law           | 7.7B     |    44.1     |   58.2   | **38.5** | 46.9           |
> | AdaptLLM-7B + Parameter Merging  | 7B       |    45.6     |   60.1   |   36.3   | 47.3_(95%)     |
> |           Twin-Merging           | 15.3B    |    45.9     |   61.3   |   36.7   | 47.9_(96%)     |
> | LLaMA-7B + Router + 3*SkillPacks | **9.1B** |    47.2     |   63.4   |   38.2   | **49.6_(99%)** |
>
> **We thank the Reviewer again for the useful comments.** We have revised the manuscript according to the Reviewer’s suggestion and response to each comment provided in the Weakness section above. **We hope that our rebuttal aligns with the reviewer’s expectations**, and we hope that the Reviewer can consider possibly giving a higher rating. Thanks.

---

> ### Comment · Reviewer_ZGH4 · 2025-11-25
>
> I thank the authors for their detailed and comprehensive response, which have addressed my concerns. I have raised my score from 6 to 8.

---

> ### Author Response · Authors · 2025-11-28
>
> Thanks so much for raising your rating from 6 to 8. We are glad that we can solve all your questions and we are very much looking forward to further discussions on our work!

---

### Official Review · Reviewer_zW7i · 2025-10-27

**Soundness:** 3
**Presentation:** 2
**Contribution:** 2
**Rating:** 4
**Confidence:** 3

**Summary:**

The paper presents a graft method, GraftLLM, that extracts the source models' capabilities as SkillPack and then dynamically assigns capabilities to the target model based on the tasks. Moreover, this method employs a combination of compression techniques, including pruning, quantization, and decomposition, to make the SkillPack compact and efficient. The router function is trained on FuseChat 2.0 to explicitly select SkillPack based on the source model and task type. The experiments show that GraftLLM outperforms the other merging or fusion methods with comparable parameters on multiple tasks.

**Strengths:**

1. Innovative router-based skill integration:
The paper introduces a novel router function that dynamically assigns SkillPacks to the target model, enabling flexible and adaptive composition of capabilities. This mechanism allows the target model to seamlessly adapt to diverse tasks without full retraining.

2. Support for heterogeneous model fusion:
Unlike most prior works that focus on homogeneous architectures, the proposed method is applicable to heterogeneous LLMs, demonstrating strong generality and robustness across models with different structures and parameterizations.

3. Strong empirical performance with compact models:
The proposed GraftLLM achieves superior performance across multiple benchmarks while maintaining a comparable parameter count to competing methods, highlighting its efficiency and effectiveness.

**Weaknesses:**

1. Terminology and clarity issues:
The paper assumes substantial prior knowledge and introduces multiple technical terms without sufficient explanation. For instance, PEFT is used in the abstract and appears frequently throughout the paper, but it is never explicitly defined. Similarly, the term modules is used ambiguously; it would help if the authors clarified that they refer to specific neural network components, such as attention layers or MLP blocks. Additionally, there are several minor typographical errors, particularly in Section 3, that should be corrected for readability.

2. Methodological complexity and unclear contribution emphasis:
While the proposed approach is empirically strong, it combines more than five techniques, which makes it conceptually complex and somewhat difficult to parse. According to the ablation study, the optimal configuration involves using pruning for the embedding and output head, SVD for MLP layers, and low-rank SVD for attention layers. However, in such a composite pipeline, the reported parameter reduction may not translate into practical efficiency—training and validation runtimes could be substantially longer than for competing methods. Although a runtime analysis is included, it lacks a direct comparison with other baselines. Moreover, if similar compression strategies (e.g., pruning, SVD) were applied to the competitors, they might outperform the proposed method. The paper would be stronger if the authors highlighted the most impactful component rather than treating all sub-techniques as equally central, as the current presentation dilutes the core contribution.

3. Insufficient explanation of the router function:
The router function appears to be the conceptual centerpiece of the paper, yet its formulation and role are not clearly articulated in the main text. A more detailed exposition—possibly with a schematic or mathematical description—would make the contribution clearer and could inspire future extensions of this idea.

**Questions:**

1. Runtime and complexity concern:
Could you provide a runtime comparison against the competing methods, including both training and validation phases? The proposed framework integrates several techniques, which may introduce substantial computational overhead and tuning difficulty despite the parameter reduction. A comparative runtime analysis would clarify the practical efficiency of the approach.

2. Clarification of the router function:
Could you elaborate on the formal definition and empirical behavior of the router function? It appears to be the key innovation that unifies the multiple techniques and enables dynamic capability transfer across heterogeneous models. A clearer explanation would help readers understand its role and significance.

---

> ### Author Response · Authors · 2025-11-25
> **Response to Reviewer zW7i (part 1/5)**
>
> Thank you for your valuable comments. We will explain your concerns point by point.
>
> ---
>
> > **Weakness 1: Writing and Terminology Improvements?**
>
> **Reply**: We acknowledge the clarity issues raised by the reviewer. In the revised version, we explicitly define PEFT and clearly specify that the term *modules* refers to attention layers, MLP blocks, and output heads. Typographical errors in Section 3 have also been corrected. These revisions aim to improve readability and reduce the assumed prior knowledge.
>
> ---
>
> > **Weakness 2：Core Contribution Clarification.**
>
> **Reply**: We thank the reviewer for the comment. To clarify, the **core contribution** of our work is the **SkillPack framework for cross-capability fusion with explicit conflict avoidance**.
>
> 1. **Modular, conflict-free knowledge:** SkillPacks decouple task-specific capabilities, preventing interference when fusing heterogeneous models.
> 2. **Heterogeneous model fusion:** Our framework safely integrates different pre-trained LLMs or architectures—something traditional parameter merging cannot achieve.
> 3. **Scalable, lightweight storage:** We are the first to show that **compressed LLM task vectors** enable scalable multi-capability integration, preserving each domain’s specialization while avoiding the degradation typically observed in direct distillation or merging.
>
> In short, the novelty lies in **safe, scalable, and modular cross-capability fusion via dynamically composable SkillPacks**.

---

> ### Author Response · Authors · 2025-11-25
> **Response to Reviewer zW7i (part 2/5)**
>
> > **Weakness 3 and Question 2： Explanation of the Router Function.**
>
> **Reply:** We thank the reviewer for the question. We have moved the router description from the appendix into the main paper (**Section 3.3**) and provided clearer architectural details in **Appendix B.3**, including the specific formula and router variant used in experiments. Our router adaptively selects the most relevant SkillPack based on task similarity or divergence (see Table 1).
>
> - **Similar tasks (e.g., Explicit/Implicit LLM Fusion):** When SkillPacks represent highly similar capabilities, their decision boundaries overlap, making top-1 routing **intrinsically ambiguous**. Routing to multiple SkillPacks (e.g., top-2) can yield noticeable performance gains. However, each SkillPack must first be decompressed and its delta applied to **reconstruct full model parameters**, meaning the inference cost **scales almost linearly with the number of SkillPacks**. For efficiency and fairness, our main experiments use **top-1 routing**, activating only the most relevant SkillPack while minimizing overhead and retaining the benefits of modular, task-specific deltas.
> - **Divergent tasks (e.g., Finance–Bio–Law LLM Fusion):** When SkillPacks are strongly differentiated, the router easily maps inputs to the correct domain expert. Here, top-1 routing suffices, and ensembling provides little benefit because the knowledge is not mutually reinforcing.
>
> Across all scenarios, the router behaves as intended: it enables **synergy** when tasks align, prevents harmful **cross-task interference** when tasks diverge, and **preserves efficiency** through selective expert activation. Experimental results consistently validate the router’s robustness in both similar-task and diverse-task fusion settings.
>
> Table 1: Analysis of router behavior and multi-SkillPack ensembling across three fusion settings.
>
> |          Methods          | Explicit LLM Fusion (*AlpacaEval 2.0*) | Implicit LLM Fusion (*10 benchmark*) | Fusion Finance+Law+Bio LLM | Inference Time |
> | :-----------------------: | :------------------------------------: | :----------------------------------: | :------------------------: | -------------- |
> |         FuseChat          |             11.63 / 16.89              |                 55.2                 |           47.33            | 1x             |
> | Grafting Top 1 SkillPack  |  16.56 **(+4.9)** / 25.42 **(+8.5)**   |           56.0 **(+0.8)**            |     49.62 **(+2.29)**      | 1.03x          |
> | Grafting Top 2 SkillPacks |      17.33 (+0.8) / 26.28 (+0.9)       |             56.3 (+0.3)              |     **49.64** (+0.02)      | 2.03x          |
> | Grafting Top 3 SkillPacks |  **17.49** (+0.2) / **26.72** (+0.4)   |           **56.4** (+0.1)            |       49.63 (-0.01)        | 3.03x          |

---

> ### Author Response · Authors · 2025-11-25
> **Response to Reviewer zW7i (part 3/5)**
>
> > **Question 1：Runtime and Complexity Discussion.**
>
> **Reply**: We thank the reviewer for raising this point. In the revised manuscript, we have added a direct runtime comparison with the main competing methods. As shown in Table 2, our approach maintains **comparable training and inference costs** to prior work, despite integrating several compression techniques. The only additional overhead comes from the **one-time compression stage**, which is slightly longer because we apply quantization during task-vector compression. However, this cost is incurred **only once** and does not affect training. More importantly, quantization **substantially reduces the storage and memory footprint of task vectors**, greatly improving the deployability and practicality of dynamic capability fusion in large models.
>
> Overall, the updated runtime results confirm that our framework delivers **significant parameter savings and stronger performance** compared with existing baselines, while still maintaining **competitive computational efficiency**. In addition, detailed per-module compression times for each model are provided in **Appendix F.3 and Table 18**.
>
> Table 2: Training and inference resource comparison between SkillPack and representative LLM fusion baselines.
>
> |                         Method                         | Model Size | Router Size | Training Cost (GPU h) | Compression Cost | Inference Overhead |
> | :----------------------------------------------------: | :--------: | :---------: | :-------------------: | :--------------: | :----------------: |
> |          Multi-Teacher Distillation (FuseLLM)          |     7B     |      –      |         132 h         |        –         |       1.00×        |
> |  Pairwise Distillation + Parameter Merging (FuseChat)  |     7B     |      –      |         132 h         |      4 min       |       1.00×        |
> | Pairwise Distillation + Dynamic Merging (Twin-Merging) |    21B     |     21M     |     132 h + 8 min     |      9 min       |       1.03×        |
> |      **Pairwise Distillation + SkillPack (Ours)**      |    9.2B    |     21M     |     132 h + 8 min     |     163 min      |       1.03×        |

---

> ### Author Response · Authors · 2025-11-25
> **Response to Reviewer zW7i (part 4/5)**
>
> > **Additional Analysis: Results on fusion domains with much larger intrinsic divergence**
>
> **Reply**: Our method is explicitly designed to **decouple conflicting task behaviors into separate SkillPacks**, preventing cross-task interference and enabling near-lossless capability fusion—even across highly distinct domains.
>
> To demonstrate this property, we conducted an additional experiment on domains with **much larger intrinsic divergence** (finance, law, and biomedicine), following the evaluation protocol of the ICLR 2024 paper *“Adapting Large Language Models to Domains via Reading Comprehension.”*
>
> We first evaluate models individually fine-tuned on each domain. As shown in Table 3 below, a model fine-tuned on one domain **suffers substantial degradation** on the others, sometimes performing worse than the base model. This confirms that traditional parameter-merging approaches cannot resolve strong **negative interference** across domains.
>
> **In contrast, our SkillPack-based fusion effectively isolates domain-specific updates and recombines them with minimal interference, achieving near-lossless multi-domain performance.**
>
> Table 3: Performance comparison between GraftLLM and representative LLM fusion baselines, evaluated across three domains: Biomedicine, Finance, and Law.
>
> |             Methods              | Params   | Biomedicine | Finance  |   Law    | Average        |
> | :------------------------------: | -------- | :---------: | :------: | :------: | -------------- |
> |             LLaMA-7B             | 7B       |    44.2     |   58.6   |   34.2   | 45.7           |
> |    AdaptLLM-7B [1]\[ICLR24\]     | **3*7B** |    47.3     |   63.4   |   38.5   | 49.7_(100%)    |
> |           LLaMA-7B-Bio           | 7.7B     |  **47.3**   |   57.9   |   34.5   | 46.6           |
> |         LLaMA-7B-Finace          | 7.7B     |    43.7     | **63.4** |   34.0   | 57.0           |
> |           LLaMA-7B-Law           | 7.7B     |    44.1     |   58.2   | **38.5** | 46.9           |
> | AdaptLLM-7B + Parameter Merging  | 7B       |    45.6     |   60.1   |   36.3   | 47.3_(95%)     |
> |           Twin-Merging           | 15.3B    |    45.9     |   61.3   |   36.7   | 47.9_(96%)     |
> | LLaMA-7B + Router + 3*SkillPacks | **9.1B** |    47.2     |   63.4   |   38.2   | **49.6_(99%)** |

---

> ### Author Response · Authors · 2025-11-25
> **Response to Reviewer zW7i (part 5/5)**
>
> > **Additional Analysis: How Do SkillPacks Prevent Cross-Task Interference During Fusion?**
>
> **Reply**:  Our method is designed to *structurally decouple* **task- and domain-specific updates** into separate SkillPacks. This modularization prevents gradient or parameter-level interference and enables stable multi-task fusion with minimal accuracy loss.
>
> To better illustrate this property, we added an additional experiment using domains with much **larger intrinsic differences**—finance, law, and biomedicine—following the setup of the ICLR 2024 paper *“Adapting LLMs to Domains via Reading Comprehension.”*
>
> **(1). Baseline Observation: Strong Domain Interference.**
>  Table 3 shows that a model fine-tuned on any single domain generalizes **poorly** to the others, sometimes performing even worse than the pretrained model. This highlights that **conventional parameter merging** cannot overcome severe negative interference across domains.
>
> **(2). Why SkillPacks Avoid This Problem?**
>  SkillPacks **isolate domain-specific parameter shifts**, preventing interference across domains during fusion. When recombined, each SkillPack contributes only its localized update, enabling near-lossless performance across all specialized domains.
>
> **(3). Clarifying the Original Experimental Setting.**
> In our explicit/implicit LLM fusion experiments, the tasks appear relatively similar because the underlying source LLMs naturally **share broad general-purpose abilities** (e.g., dialogue, reasoning, basic math). As a result, their behaviors naturally overlap, and the effective **domain gap is smaller**—not because the task design is narrow, but because the models themselves share strong general capabilities. This **reduces** cross-task interference, which explains why the improvements from SkillPack composition are **modest** in this setting.
>
> **(4). Takeaway.**
>  The extended experiment confirms a clear trend: **as domain differences increase and task interference becomes more substantial, the benefits of SkillPack-based fusion grow significantly.**
>
> ---
>
> > **Additional Analysis: Qualitative case study of router behavior.**
>
> In addition, we present qualitative cases showing how the router handles inputs that naturally mix biomedical, financial, and legal terminology. For each input, we report the selected SkillPack and the type of domain knowledge it contains. As shown in Table 4 below, the router reliably selects the SkillPack aligned with the **dominant semantic cue**, and the corresponding SkillPack indeed captures coherent **domain-specific knowledge** (e.g., dose–response reasoning in Biomed, regulatory interpretation in Law). This provides a qualitative understanding of what each SkillPack encodes and how the router manages mixed-domain queries.
>
> Table 4: Qualitative Case Studies on SkillPack Knowledge.
>
> | Input Example                                                | Router (Top-1) | Knowledge Captured in SkillPack                              |
> | ------------------------------------------------------------ | ------------------------- | ------------------------------------------------------------ |
> | 1. “Does increasing the dose improve overall return under the current treatment policy?” | Biomed                    | Captures clinical reasoning such as dose–response relationships, treatment efficacy, and pharmacological effects. |
> | 2. “What regulatory constraints apply when reporting adverse outcomes that may affect financial liability?” | Law                       | Encodes legal concepts including regulatory compliance, reporting requirements, and liability interpretation. |
> | 3. “Evaluate whether the compound shows significant impact on long-term yield.” | Biomed                    | Stores biochemical and pharmacological knowledge regarding compound effects, biological impact, and experimental outcomes. |
>
> ---
>
> **We thank the Reviewer again for the useful comments.** We have revised the manuscript according to the Reviewer’s suggestion and response to each comment provided in the Weakness section above. **We hope that our rebuttal aligns with the reviewer’s expectations**, and we hope that the Reviewer can consider possibly giving a higher rating. Thanks.

---

> > ### Comment · Reviewer_zW7i · 2025-11-25
> >
> > Thank you for the response. Some of my concerns have been resolved, but my concern regarding Weakness 2 remains. Since the paper adopts more than five techniques, the authors should provide an ablation study to isolate the effect of each component (for example, how important is the router regarding the other techniques adopted?). This would help readers understand the core intuition and make it easier to build on the SkillPack framework. For instance, among pruning, SVD, quantization, and the router, which contributes the most to the fusion performance? Is the router the main driver of accuracy, while the others primarily improve efficiency? Could you provide an extended ablation study to show how important the router function is? Clarifying these points would significantly strengthen the narrative and technical takeaway of the paper.
> >
> > Lastly, I noticed that the final two parts of the response appear to address points I did not raise. The authors may want to verify whether those responses were placed in the correct section.

---

> > > ### Author Response · Authors · 2025-11-26
> > > **Second Response to Reviewer zW7i (part 1/3)**
> > >
> > > ### We sincerely appreciate the reviewer’s response and the valuable suggestions. Your feedback is highly helpful for refining both the technical contributions and experimental validation of GraftLLM. We address your concerns point by point as follows:
> > >
> > > ---
> > >
> > > **1. Clarifying the Core Contribution.**
> > >  To help readers better understand the core intuition of our method and facilitate future work built upon the SkillPack framework, we provide a systematic comparison against existing approaches in **Tables 1, 2, and 3** below, and offer a comprehensive technical analysis in original **Appendix A**. Our key contribution is replacing traditional **static parameter merging** with a **router + delta-compression** mechanism in multi-teacher LLM distillation, thereby avoiding task conflicts among heterogeneous sources.
> > >
> > > The motivation behind GraftLLM is as follows: in practical applications, **multi-teacher distillation$^{[1]}$** often becomes extremely difficult to train due to severe **data heterogeneity**. FuseChat-2$^{[2]}$ attempted to address this by combining pair-wise distillation with parameter merging to mitigate incompatibility among heterogeneous LLMs. However, because parameter conflicts remain fundamentally unavoidable, **the improvements were limited**. Consequently, FuseChat-3$^{[3]}$ **abandoned parameter merging entirely and returned to pure multi-teacher distillation**.
> > >
> > > In contrast, our work **reintroduces dynamic (non-static) model fusion** into the heterogeneous LLM setting, making model merging **feasible again** and **restoring confidence in the distillation + merging paradigm**. Model merging has been widely adopted in recent years due to its simplicity, efficiency, and ease of use, and we hope that dynamic model merging can open a new direction for multi-LLM distillation.
> > >
> > > As modern LLMs rapidly expand toward multimodal and multi-capability scenarios, **data heterogeneity and cross-capability transfer** become increasingly challenging. GraftLLM provides a new dynamic fusion solution to address this challenge.
> > >
> > > Table 1: Comparison of key components used by GraftLLM and representative prior methods.  Router = dynamic task routing; Delta Compression = compressed task vector technique.  ✓ indicates the component is used; × indicates it is not.
> > >
> > > |                    Method                     | Distillation | Parameter Merging | Router | Delta Compression |
> > > | :-------------------------------------------: | :----------: | :---------------: | :----: | :---------------: |
> > > |      FuseLLM$^{[1]}$, FuseChat-3$^{[3]}$      |      ✔️       |         ❌         |   ❌    |         ❌         |
> > > |    FuseChat-2$^{[2]}$, InfiFusion$^{[4]}$     |      ✔️       |         ✔️         |   ❌    |         ❌         |
> > > |    LoRAHub$^{[5]}$, LoRA-MoE$^{[6]}$, etc.    |      ❌       |         ❌         |   ✔️    |         ❌         |
> > > |    TALL-Mask$^{[7]}$, EMR-Merging$^{[8]}$     |      ❌       |         ✔️         |   ✔️    |         ❌         |
> > > |             Twin-Merging$^{[9]}$              |      ❌       |         ✔️         |   ✔️    |         ✔️         |
> > > |  DARE$^{[10]}$, TIES-Merging$^{[11]}$, etc.   |      ❌       |         ✔️         |   ❌    |         ✔️         |
> > > | BitaDelta$^{[12]}$, Delta-Come$^{[13]}$, etc. |      ❌       |         ❌         |   ❌    |         ✔️         |
> > > |              **GraftLLM (Ours)**              |      ✔️       |         ❌         |   ✔️    |         ✔️         |

---

> ### Author Response · Authors · 2025-11-26
> **Second Response to Reviewer zW7i (part 2/3)**
>
> Table 2: Comparison of heterogeneous LLM fusion approaches. GraftLLM uniquely combines pair-wise distillation with preference optimization and dynamic, modular fusion for scalable cross-capability integration.
>
> |                 Method                 | Distillation  | Preference Optimization | Model Merging Approach |
> | :------------------------------------: | :-----------: | :---------------------: | :--------------------: |
> |            FuseLLM$^{[1]}$,            | Multi-teacher |            ❌            |           ❌            |
> | FuseChat-2$^{[2]}$, InfiFusion$^{[4]}$ |   Pair-wise   |            ❌            |     Static Merging     |
> |           FuseChat-3$^{[3]}$           | Multi-teacher |            ✔️            |           ❌            |
> |          **GraftLLM (Ours)**           |   Pair-wise   |            ✔️            |  Dynamic and Modular   |
>
> Table 3: Comparison of delta-compression techniques, where delta-compression refers to methods that store task-specific **parameter updates** in a compact form. Columns indicate whether a method employs delta pruning, SVD, or quantization strategies. **GraftLLM is the first to apply delta-compression techniques for scalable, modular, and conflict-free LLM fusion.**
>
> |                  Delta-Compression Methods                   | Delta Pruning | Delta SVD | Delta Quantization | Module Adaptive |
> | :----------------------------------------------------------: | :-----------: | :-------: | :----------------: | :-------------: |
> | DARE$^{[10]}$, TIES-Merging$^{[11]}$, NPS-Pruning$^{[16]}$, ect. |       ✔️       |     ❌     |         ❌          |        ❌        |
> |  Twin-Merging$^{[9]}$, KnOTS$^{[15]}$, D$^{2}$-MoE$^{[14]}$  |       ❌       |     ✔️     |         ❌          |        ❌        |
> |                      BitaDelta$^{[12]}$                      |       ❌       |     ❌     |         ✔️          |        ❌        |
> |             ImPart$^{[17]}$, Delta-Come$^{[13]}$             |       ❌       |     ✔️     |         ✔️          |        ❌        |
> |                     **GraftLLM (Ours)**                      |       ✔️       |     ✔️     |         ✔️          |        ✔️        |
>
> References: \
>    [1] Knowledge Fusion of Large Language Models. (ICLR 2024)  \
>    [2] Fusechat: Knowledge fusion of chat models. (EMNLP 2025)  \
>    [3] FuseChat-3.0: Preference Optimization Meets Heterogeneous Model Fusion. (ICLR 2025) \
>    [4] InfiFusion: A Unified Framework for Enhanced Cross-Model Reasoning via LLM Fusion. (NeurIPS 2025) \
>    [5] Lorahub: Efficient cross-task generalization via dynamic lora composition. (COLM 2024) \
>    [6] Self-moe: Towards compositional large language models with self-specialized experts. (ICLR 2025) \
>    [7] Localizing task information for improved model merging and compression. (ICML24) \
>    [8] Emr-merging: Tuning-free high-performance model merging. (NeurIPS 2024)  \
>    [9] Twin-merging: Dynamic integration of modular expertise in model merging. (NeurIPS 2024)  \
>    [10] Language models are super mario: Absorbing abilities from homologous models as a free lunch. (ICML24) \
>    [11] Ties-merging: Resolving interference when merging models. (NeurIPS 2024)  \
>    [12] Bitdelta: Your fine-tune may only be worth one bit. (NeurIPS 2024)  \
>    [13] Delta-CoMe: Training-Free Delta-Compression with Mixed-Precision for Large Language Models. (NeurIPS 2024)  \
>    [14] Delta Decompression for MoE-based LLMs Compression. (ICML25) \
>    [15] Model merging with SVD to tie the Knots. (ICLR 2025) \
>    [16] Neural Parameter Search for Slimmer Fine-Tuned Models. (ACL 2025) \
>    [17] ImPart: Importance-Aware Delta-Sparsification for Improved Model Compression and Merging in LLMs. (ACL 2025)

---

> ### Author Response · Authors · 2025-11-26
> **Second Response to Reviewer zW7i (part 3/3)**
>
> **2. Clarification on Comparative Experiments.**
>  We have provided comprehensive comparative experiments in the original paper. We use the *Router* to select the appropriate expert to recover full-model fine-tuning performance, and we apply *Delta Compression* to improve the efficiency of representing parameter updates. We provide separate ablation analyses for both components as follows:
>
> **(1) Ablation Study of the Router**
>  We discuss two scenarios separately:
>
> - **Explicit LLM Fusion:** Since different source models possess similar general conversational capabilities, we train the router on per-sample performance of each source model in the training set. Appendix **B.3** and Table **6** (Table **8** in the revised version) analyze the **effectiveness of the router** in explicit fusion. Results show that the router has a significant impact on performance, and **more training data** improves both routing quality and fusion outcomes.
> - **Implicit Fusion & Bio–Finance–Law Cross-Domain LLM Fusion:** Here, we use an **ideal router**, manually selecting the most appropriate SkillPack based on task type. In the revised version, we will further include results demonstrating the effect of **router training** in these scenarios.
>
> **(2) Ablation Study of Delta Compression**
>  We also discuss two aspects:
>
> - **Comparison with existing delta-compression methods:** In Section **5.1** and Figures **7–8** of the original paper, we comprehensively compare our proposed strategy with delta-pruning, delta-SVD, delta-quantization, and other methods, while analyzing performance differences under SFT and DPO settings.
> - **Component analysis of the module-aware adaptive strategy:** In Section **6.1** and Table **4** (Table **5** in the revised version), we provide an ablation of each component within the strategy. Even though compression primarily improves parameter efficiency, we report the accuracy impact of different strategies under **the same parameter budget**.
>    We further extend this comparison in Appendix **B.1** and Table **5** (Table **7** in the revised version). Figures **5–6** in Section **3.2** of the original paper also illustrate the effects of different strategies on various modules. Additionally, Appendix **B.2** and Figure **10** in the original manuscript present the performance trends across **compression hyperparameters** under our proposed module-aware compression strategy.
>
> ---
>
> **Lastly, regarding the final two points in our first response, thank you for the reminder — they were not mistakes. We intentionally provided additional analysis to ensure clarity and completeness, as we greatly value the opportunity to engage with the reviewers. We sincerely hope that the second response addresses your remained concerns and welcome any further discussion on topics related to GraftLLM.**

---

> > ### Comment · Reviewer_zW7i · 2025-11-28
> >
> > Thank you for the responses. My questions are resolved. I would raise the score.

---

> > > ### Author Response · Authors · 2025-11-28
> > >
> > > Thanks so much for raising yor rating from 4 to 6. We are glad that we can solve all your questions and we are very much looking forward to further discussions on our work !

---

### Official Review · Reviewer_HvCk · 2025-10-30

**Soundness:** 2
**Presentation:** 2
**Contribution:** 4
**Rating:** 2
**Confidence:** 3

**Summary:**

The paper addresses the problem of LLMs merging. The paper proposed an approach which consists of storing source model capabilities in a target model + SkillPack format. According to the paper, this approach preserves general capabilities, reduces parameter conflicts, and supports forget-free continual learning and model fusion. According to the paper, the approach proposed outperforms existing techniques.

**Strengths:**

The paper addresses a well known problem when developing or adapting LLMs. Instead of developing a completely new model, the goal is to merge existing models. The paper proposes a new approach which  preserves general capabilities, reduces parameter conflicts, and supports forget-free continual learning.

**Weaknesses:**

The main issue with the paper is self contained which makes its review difficult. The paper used several key technical terms such as “cross-capability transfer”, “SkillPack”, MLP, etc. without providing the meaning, which can make the paper very difficult to understand by newcomers.

The problem addressed by the paper is not clear and there is a confusion in the paper contribution:
- The paper is addressing the problem when models are structurally identical
- What is the difference between heterogeneous models (line 58) and structural different model (line 60) that the paper wants to solve? These terms lead to confusion. The paper should clarify all the terms used.

The methodology is not understandable, which did not make the reproducibility easy. Several capital terms are not clearly defined. Figure 4 is not well explained. Section 3.2 is not clear

Some description of key terms (line 267) are provided in additional materials. However, the paper should be self contained.

Experimentation environment, hyper-parameter, etc. are not provided in the experimentation section.

The evaluation of the time complexity and difficulties to integrate the models are not provided.
- Line 26: for heterogeneous heterogeneous LLM fusion –> there is a redundancy of the term “heterogeneous”
- Missing experimentation and results summary in the abstract. The paper could present to which approach the proposed approach was compared to.
- SkillPack, cross-capability transfer should be defined before first use in the introduction and abstract
- In the abstract, the paper could introduce the experiments, present to which baseline their approach was compared to and summarize some results.

- Line 106: “SFT and DPO” should be defined before first use
- Line 112-113 is not clear.
- The figures presented in the paper should be accompanied by a description text which makes its understanding easy.
- The definition of SkillPack in line 215 is too short and not understandable for such a technical term. The paper should provide a clear explanation of what SkillPack is (with examples).
- The ablation may provide the evaluation (costs in time of the different approaches) and  the integration difficulties.

**Questions:**

- What is Modular Skill Pack?

---

> ### Author Response · Authors · 2025-11-25
>
> > Thank you for the detailed comments. We address each concern below and clarify several misunderstandings.
> >
> > ------
>
> ##### **Reply**:  **1. “The paper is not self-contained; key terms are not defined.”**
>
> We appreciate the suggestion. In the revised version, we have added **clear definitions** for *SkillPack*, *cross-capability transfer*, and other technical terms at their first appearance.
> However, some of these concepts were already introduced in the original version, and the confusion may stem from overlooking the corresponding sections.
>
> ------
>
> ##### **2. “The problem addressed by the paper is unclear; confusion between heterogeneous and structurally different models.”**
>
> This is a misunderstanding.
>
> - Prior fusion methods target **structurally identical** models.
> - Our work explicitly focuses on **structurally different / heterogeneous models**, which are equivalent in our context.
>
> ------
>
> ##### **3. “The methodology, Figure 4, and Section 3.2 are not understandable.”**
>
> Figure 4 is the main pipeline illustration. In the original submission, each component of the figure was explained in Section 3, and the caption detailed its structure. The figure is clear when viewed together with its referenced paragraphs.
>
> Nonetheless, we have further refined the caption and added explicit cross-references in the revised version.
>
> ------
>
> ##### **4. “Experimentation environment and hyper-parameters are missing.”**
>
> These details were intentionally placed in the appendix—following common practice for large LLM papers—to keep the main text focused on analysis and findings.
>
> All training details and hyper-parameters are already provided in **Appendix D**, and we added clearer pointers in the main text.
>
> ------
>
> ##### **5. “Time complexity and integration difficulty are not provided.”**
>
> Time complexity analysis already appears in **Appendix F.3** in the original submission. We expanded this discussion in the revised version.
>  Regarding integration difficulty, our method is intentionally lightweight, and we added a concise summary explaining this design.
>
> ------
>
> ##### **6. Typos and minor clarity suggestions.**
>
> - The duplicated word “heterogeneous” has been corrected.
> - Definitions for SFT and DPO are now provided upon first use.
>
> ------
>
> ##### **7. “Missing results summary and baseline specification in the abstract.”**
>
> The original abstract already summarized our experimental settings and key results.
>  Given that our work spans three scenarios (knowledge transfer, knowledge fusion, and forget-free learning), each with different baselines, listing all of them in the abstract would make it overly long and unfocused. We therefore kept the concise structure and left baseline details in the experiment sections.
>
> ------
>
> ##### **8. “SkillPack definition is too short.”**
>
> SkillPack was introduced in the Introduction, but we agree that more detail improves clarity.
>  We expanded the definition and added an illustrative example.
>
> ------
>
> ##### **9. Ablation: cost evaluation and integration difficulty.**
>
> These points were already addressed in the appendix and further clarified in the revision.
>
> ---
>
> **We thank the Reviewer again for the comments.** We have revised the manuscript according to the Reviewer’s suggestion and response to each comment provided in the Weakness section above. **We hope that our rebuttal aligns with the reviewer’s expectations**, and we hope that the Reviewer can consider possibly giving a higher rating. Thanks.

---

> > ### Comment · Reviewer_HvCk · 2025-11-28
> >
> > The responses to my comments are addressed
> >
> > Thanks to the authors

---

> > > ### Author Response · Authors · 2025-12-02
> > >
> > > The meaning of this response is somewhat ambiguous. We assume that the reviewer intended to express that “my comments have been adequately addressed,” as no further concerns or questions were raised at the time the reviewer provided this comment.

---

> ### Author Response · Authors · 2025-11-28
>
> We are glad that we can solve all your questions and could you please consider a higher rating since you have no further concerns.

---

### Official Review · Reviewer_vnrD · 2025-10-30

**Soundness:** 3
**Presentation:** 2
**Contribution:** 2
**Rating:** 4
**Confidence:** 4

**Summary:**

This paper introduces GraftLLM, a framework for transferring capabilities across heterogeneous large language models by creating and composing modular "SkillPacks." The method involves fine-tuning a target model, extracting the parameter delta, applying a module-aware compression strategy, and composing these SkillPacks using a routing mechanism. The approach is evaluated on tasks including pairwise knowledge transfer, heterogeneous model fusion, and forget-free continual learning.

**Strengths:**

The paper tackles the highly relevant and challenging problem of fusing knowledge across heterogeneous LLMs. The proposed "module-aware adaptive compression" strategy is an intuitive and empirically effective contribution for creating compact, transferable knowledge modules. The experimental results consistently demonstrate strong performance across multiple benchmarks, outperforming several baselines in knowledge fusion.

**Weaknesses:**

The technical contribution of this work appears to be incremental. The proposed pipeline—distilling knowledge, calculating a delta, compressing it, and then composing these modules—is conceptually very similar to existing frameworks like FuseChat (pairwise distillation followed by merging) and LoraHub (dynamic composition of LoRA modules). The main novelty seems to lie in the "module-aware adaptive compression" strategy, but the overall framework feels like a combination of established techniques rather than a new approach. The paper would be stronger if it more clearly articulated its core conceptual departure from these related works.

Furthermore, the analysis provided is limited and somewhat misaligned with the paper's main theme of knowledge fusion. The primary analysis focuses on ablation studies of the compression strategy (Table 4) and its impact on performance loss. While this validates a component of the method, it does little to illuminate the dynamics of the fusion process itself. An analysis of inter-SkillPack interference, the behavior of the router on ambiguous inputs, or a qualitative look at what knowledge is captured within a SkillPack would have been more insightful for a paper centered on fusion.

The paper is difficult to follow in several key areas, with missing details and confusing notation that undermine the reader's confidence in the methodology and results.

1.  **Inconsistent Notation**: The notation for the base model is inconsistent. Equation (10) uses `θ_tgt`, while the description on line 270 uses `θ_target`. This should be standardized for clarity.

2.  **Missing Quantization Details**: The method utilizes GPTQ for quantization, which typically requires a calibration set to determine quantization parameters. The paper fails to mention what calibration data was used, how it was selected, or its size. This is a critical detail for reproducibility.

3.  **Confusing Formulation of the Router**: Equations (10) and (11) are nearly identical, yet they are meant to describe different scenarios. This implies that the router `R` and the SkillPacks themselves might function differently in each context, which is highly confusing. The paper should explicitly state this difference. Moreover, the appendix reveals that two different types of routers are used—one trained as a classifier and another based on manual, task-type assignment. This crucial distinction is not clearly made in the main paper. Important details about the trained router's architecture, training data, and hyperparameters are also missing.

4.  **Misleading Claims about Parameter Size**: The paper claims its method is efficient, yet in Table 1, the parameter count for "Routed GraftLLM (Ours)" is 9.2B, a 28% increase over the 7B base model. This increase can hardly be considered minor. Why does a method centered on *compression* result in a significantly larger final model?

5.  **Flawed Comparison in Continual Learning**: Table 3 evaluates forget-free learning. The setup is problematic for two reasons. First, it is missing some critical results: the performance of the LLaMA3 model after being fine-tuned sequentially on different stages. Second, the paper states that the GraftLLM router for this experiment is a manual one based on "task-type." If the model is explicitly told which task it is performing (e.g., "this is a math problem, load the math SkillPack"), then the comparison to other methods that do not receive this oracle information is unfair.

**Questions:**

See weaknesses

---

> ### Author Response · Authors · 2025-11-25
> **Response to Reviewer vnrD (part 1/5)**
>
> Thank you for your valuable comments. We will explain your concerns point by point.
>
> ---
>
> > **Question 1: How Is Our Method Conceptually Different from FuseChat and LoRAHub?** \
>
> **Reply**: Thank you for the question. Our method differs fundamentally from both FuseChat and LoRAHub in motivation, design, and capability aggregation.
>
> **(1) Difference from FuseChat (pairwise distillation + parameter merging).**
>
>  **Conflict avoidance:** FuseChat merges parameter deltas, which can introduce interference. Our SkillPacks preserve each domain capability independently, preventing cross-task conflicts.
>  **Robustness:** Delta merging struggles with divergent models; our method avoids merging entirely.
>  **Balanced retention:** SkillPacks maintain high performance across heterogeneous domains.
>
> **Summary:** FuseChat merges parameters; we decouple capabilities for modular, conflict-free integration.
>
> **(2) Difference from LoRAHub (LoRA-based composition).**
>
> **Full capability preservation:** LoRA adapters are low-rank and may lose fine-grained knowledge. We extract full model capabilities before compression.
> **Post-hoc adaptive compression:** Our compression reduces storage while retaining essential knowledge, enabling better cross-domain fusion than LoRA-based methods.
>
> **Summary:** LoRAHub combines low-rank LoRA adapters, which inherently limits the achievable performance, whereas we preserve **full model capabilities** before applying post-hoc compression, enabling conflict-free fusion with higher performance ceilings.
>
> ---
>
> To help readers better understand the core intuition of our method and facilitate future work built upon the SkillPack framework, we provide a systematic comparison against existing approaches in **Tables 1, 2, and 3** below, and offer a comprehensive technical analysis in original **Appendix A**. Our key contribution is replacing traditional **static parameter merging** with a **router + delta-compression** mechanism in multi-teacher LLM distillation, thereby avoiding task conflicts among heterogeneous sources.
>
> Table 1: Comparison of key components used by GraftLLM and representative prior methods.  Router = dynamic task routing; Delta Compression = compressed task vector technique.  ✓ indicates the component is used; × indicates it is not.
>
> |                    Method                     | Distillation | Parameter Merging | Router | Delta Compression |
> | :-------------------------------------------: | :----------: | :---------------: | :----: | :---------------: |
> |      FuseLLM$^{[1]}$, FuseChat-3$^{[3]}$      |      ✔️       |         ❌         |   ❌    |         ❌         |
> |    FuseChat-2$^{[2]}$, InfiFusion$^{[4]}$     |      ✔️       |         ✔️         |   ❌    |         ❌         |
> |    LoRAHub$^{[5]}$, LoRA-MoE$^{[6]}$, etc.    |      ❌       |         ❌         |   ✔️    |         ❌         |
> |    TALL-Mask$^{[7]}$, EMR-Merging$^{[8]}$     |      ❌       |         ✔️         |   ✔️    |         ❌         |
> |             Twin-Merging$^{[9]}$              |      ❌       |         ✔️         |   ✔️    |         ✔️         |
> |  DARE$^{[10]}$, TIES-Merging$^{[11]}$, etc.   |      ❌       |         ✔️         |   ❌    |         ✔️         |
> | BitaDelta$^{[12]}$, Delta-Come$^{[13]}$, etc. |      ❌       |         ❌         |   ❌    |         ✔️         |
> |              **GraftLLM (Ours)**              |      ✔️       |         ❌         |   ✔️    |         ✔️         |
>
> ---
>
> Table 2: Comparison of heterogeneous LLM fusion approaches. GraftLLM uniquely combines pair-wise distillation with preference optimization and dynamic, modular fusion for scalable cross-capability integration.
> |                 Method                 | Distillation  | Preference Optimization | Model Merging Approach |
> | :------------------------------------: | :-----------: | :---------------------: | :--------------------: |
> |            FuseLLM$^{[1]}$,            | Multi-teacher |            ❌            |           ❌            |
> | FuseChat-2$^{[2]}$, InfiFusion$^{[4]}$ |   Pair-wise   |            ❌            |     Static Merging     |
> |           FuseChat-3$^{[3]}$           | Multi-teacher |            ✔️            |           ❌            |
> |          **GraftLLM (Ours)**           |   Pair-wise   |            ✔️            |  Dynamic and Modular   |

---

> ### Author Response · Authors · 2025-11-25
> **Response to Reviewer vnrD (part 2/5)**
>
> > **Question 2：The Core Contribution Beyond Adaptive Compression.**
>
> **Reply**: We appreciate the question. While adaptive compression reduces storage, it is **not the core contribution**. The key innovation is a new **cross-capability fusion framework** that explicitly addresses *knowledge conflict and forgetting* when integrating heterogeneous LLMs.
>
> - **SkillPack representation:** Decouples domain-specific knowledge, preventing interference and enabling modular, near-lossless multi-task fusion.
> - **Scalable integration:** Compressed task vectors support multi-capability fusion without degrading domain specialization.
> - **Conflict-free aggregation:** Unlike parameter-merging methods, our approach preserves individual domain performance while allowing efficient combination.
>
> **Summary:** Our work introduces a principled paradigm for conflict-free, modular cross-capability fusion, fundamentally different from prior merging-based approaches.
>
> We are also **the first to demonstrate** that **compressed LLM task vectors** can support *scalable*, *multi-capability* integration, **maintaining each domain’s specialization** while avoiding the degradation typically observed in direct distillation or merging.
>
> Table 3: Comparison of delta-compression techniques, where delta-compression refers to methods that store task-specific **parameter updates** in a compact form. Columns indicate whether a method employs delta pruning, SVD, or quantization strategies. **GraftLLM is the first to apply delta-compression techniques for LLM fusion, it is scalable, modular, and conflict-free.**
> |                  Delta-Compression Methods                   | Delta Pruning | Delta SVD | Delta Quantization | Module Adaptive |
> | :----------------------------------------------------------: | :-----------: | :-------: | :----------------: | :-------------: |
> | DARE$^{[10]}$, TIES-Merging$^{[11]}$, NPS-Prunin$^{[16]}$g, ect. |       ✔️       |     ❌     |         ❌          |        ❌        |
> |  Twin-Merging$^{[9]}$, KnOTS$^{[15]}$, D$^{2}$-MoE$^{[14]}$  |       ❌       |     ✔️     |         ❌          |        ❌        |
> |                      BitaDelta$^{[12]}$                      |       ❌       |     ❌     |         ✔️          |        ❌        |
> |             ImPart$^{[17]}$, Delta-Come$^{[13]}$             |       ❌       |     ✔️     |         ✔️          |        ❌        |
> |                     **GraftLLM (Ours)**                      |       ✔️       |     ✔️     |         ✔️          |        ✔️        |
>
> ---
>
> Additionally, we provided a comprehensive Ablation Study of Delta Compression from two perspectives:
>
> - **Comparison with existing delta-compression methods:** In Section **5.1** and Figures **7–8** of the original paper, we comprehensively compare our proposed strategy with delta-pruning, delta-SVD, delta-quantization, and other methods, while analyzing performance differences under SFT and DPO settings.
> - **Component analysis of the module-aware adaptive strategy:** In Section **6.1** and Table **4** (Table **5** in the revised version), we provide an ablation of each component within the strategy. Even though compression primarily improves efficiency, we report the accuracy impact of different strategies under the same parameter budget.
>    We further extend this comparison in Appendix **B.1** and Table **5** (Table **7** in the revised version). Figures **5–6** in Section **3.2** of the original paper also illustrate the effects of different strategies on various modules.
>
> References: \
>    [1] Knowledge Fusion of Large Language Models. (ICLR 2024)  \
>    [2] Fusechat: Knowledge fusion of chat models. (EMNLP 2025)  \
>    [3] Preference Optimization Meets Heterogeneous Model Fusion. (ICLR 2025) \
>    [4] InfiFusion: A Unified Framework for Enhanced Cross-Model Reasoning via LLM Fusion. (NeurIPS 2025) \
>    [5] Lorahub: Efficient cross-task generalization via dynamic lora composition. (COLM 2024) \
>    [6] Self-moe: Towards compositional large language models with self-specialized experts. (ICLR 2025) \
>    [7] Localizing task information for improved model merging and compression. (ICML24) \
>    [8] Emr-merging: Tuning-free high-performance model merging. (NeurIPS 2024)  \
>    [9] Twin-merging: Dynamic integration of modular expertise in model merging. (NeurIPS 2024)  \
>    [10] Language models are super mario. (ICML24) \
>    [11] Ties-merging: Resolving interference when merging models. (NeurIPS 2024)  \
>    [12] Bitdelta: Your fine-tune may only be worth one bit. (NeurIPS 2024)  \
>    [13] Delta-CoMe: Training-Free Delta-Compression with Mixed-Precision for Large Language Models. (NeurIPS 2024)  \
>    [14] Delta Decompression for MoE-based LLMs Compression. (ICML25) \
>    [15] Model merging with SVD to tie the Knots. (ICLR 2025) \
>    [16] Neural Parameter Search for Slimmer Fine-Tuned Models. (ACL 2025) \
>    [17] ImPart: Importance-Aware Delta-Sparsification for Improved Model Compression and Merging in LLMs. (ACL 2025)

---

> ### Author Response · Authors · 2025-11-25
> **Response to Reviewer vnrD (part 3/5)**
>
> > **Question 3：How Do SkillPacks Prevent Cross-Task Interference During Fusion?**
>
> **Reply**:  Our method is designed to *structurally decouple* **task- and domain-specific updates** into separate SkillPacks. This modularization prevents gradient or parameter-level interference and enables stable multi-task fusion with minimal accuracy loss.
>
> To better illustrate this property, we added an additional experiment using domains with much **larger intrinsic differences**—finance, law, and biomedicine—following the setup of the ICLR 2024 paper *“Adapting LLMs to Domains via Reading Comprehension.”*
>
> **(1). Baseline Observation: Strong Domain Interference.**
>  Table 4 shows that a model fine-tuned on any single domain generalizes **poorly** to the others, sometimes performing even worse than the pretrained model. This highlights that **conventional parameter merging** cannot overcome severe negative interference across domains.
>
> **(2). Why SkillPacks Avoid This Problem?**
>  SkillPacks **isolate domain-specific parameter shifts**, preventing interference across domains during fusion. When recombined, each SkillPack contributes only its localized update, enabling near-lossless performance across all specialized domains.
>
> **(3). Clarifying the Original Experimental Setting.**
> In our explicit/implicit LLM fusion experiments, the tasks appear relatively similar because the underlying source LLMs naturally **share broad general-purpose abilities** (e.g., dialogue, reasoning, basic math). As a result, their behaviors naturally overlap, and the effective **domain gap is smaller**—not because the task design is narrow, but because the models themselves share strong general capabilities. This **reduces** cross-task interference, which explains why the improvements from SkillPack composition are **modest** in this setting.
>
> **(4). Takeaway.**
>  The extended experiment confirms a clear trend: **as domain differences increase and task interference becomes more substantial, the benefits of SkillPack-based fusion grow significantly.**
>
> Table 4: Performance comparison between GraftLLM and representative LLM fusion baselines, evaluated across three domains: Biomedicine, Finance, and Law.
>
> |             Methods              | Params   | Biomedicine | Finance  |   Law    | Average        |
> | :------------------------------: | -------- | :---------: | :------: | :------: | -------------- |
> |             LLaMA-7B             | 7B       |    44.2     |   58.6   |   34.2   | 45.7           |
> |    AdaptLLM-7B [1]\[ICLR24\]     | **3*7B** |    47.3     |   63.4   |   38.5   | 49.7_(100%)    |
> |           LLaMA-7B-Bio           | 7.7B     |  **47.3**   |   57.9   |   34.5   | 46.6           |
> |         LLaMA-7B-Finace          | 7.7B     |    43.7     | **63.4** |   34.0   | 57.0           |
> |           LLaMA-7B-Law           | 7.7B     |    44.1     |   58.2   | **38.5** | 46.9           |
> | AdaptLLM-7B + Parameter Merging  | 7B       |    45.6     |   60.1   |   36.3   | 47.3_(95%)     |
> | LLaMA-7B + Router + 3*SkillPacks | **9.1B** |    47.2     |   63.4   |   38.2   | **49.6_(99%)** |

---

> ### Author Response · Authors · 2025-11-25
> **Response to Reviewer vnrD (part 4/5)**
>
> > **Question 4：Router Behavior and Qualitative Case Studies on SkillPack Knowledge**.
>
> **Reply**: The router’s behavior indeed varies depending on task similarity, and we have clarified this distinction in the revised manuscript.
>
> **(1) When tasks are highly similar (Explicit / Implicit LLM Fusion settings).**
> When SkillPacks represent highly similar capabilities, their decision boundaries overlap, making top-1 routing **intrinsically ambiguous**. Routing to multiple SkillPacks (e.g., top-2) can yield noticeable performance gains. However, each SkillPack must first be decompressed and its delta applied to **reconstruct full model parameters**, meaning the inference cost **scales almost linearly with the number of SkillPacks**. For efficiency and fairness, our main experiments use **top-1 routing**, activating only the most relevant SkillPack while minimizing overhead and retaining the benefits of modular, task-specific deltas.
>
> **(2) When tasks are clearly differentiated (Finance–Bio–Law settings).**
> When tasks are more distinct, the router can easily learn the optimal mapping from input patterns to source capabilities. In these settings, **top-1 routing is sufficient**, as the router naturally emphasizes the most relevant expert model. Cross-expert synergy plays a smaller role because the knowledge from different tasks does not strongly reinforce each other. This behavior is consistent with the trends observed in **Table 5 and Section 6.3**.
>
> Table 5: Analysis of router behavior and multi-SkillPack ensembling across three fusion settings.
> |          Methods          | Explicit LLM Fusion (*AlpacaEval 2.0*) | Implicit LLM Fusion (*10 benchmark*) | Fusion Finance+Law+Bio LLM | Inference Time |
> | :-----------------------: | :------------------------------------: | :----------------------------------: | :------------------------: | -------------- |
> |         FuseChat          |             11.63 / 16.89              |                 55.2                 |           47.33            | 1x             |
> | Grafting Top 1 SkillPack  |  16.56 **(+4.9)** / 25.42 **(+8.5)**   |           56.0 **(+0.8)**            |     49.62 **(+2.29)**      | 1.03x          |
> | Grafting Top 2 SkillPacks |      17.33 (+0.8) / 26.28 (+0.9)       |             56.3 (+0.3)              |     **49.64** (+0.02)      | 2.03x          |
> | Grafting Top 3 SkillPacks |  **17.49** (+0.2) / **26.72** (+0.4)   |           **56.4** (+0.1)            |       49.63 (-0.01)        | 3.03x          |
>
>  We have provided comprehensive comparative experiments in the original paper on **Ablation Study of the Router.** \
>  We discuss two scenarios separately:
>
> - **Explicit LLM Fusion:** Since different source models possess similar general conversational capabilities, we train the router on **per-sample performance of each source model** in the training set. Appendix **B.3** and Table **6** (Table **8** in the revised version) analyze the effectiveness of the router in explicit fusion. Results show that the router has a significant impact on performance, and more training data improves both routing quality and fusion outcomes.
> - **Implicit Fusion & Bio–Finance–Law LLM Fusion:** Here, we use an **ideal router**, manually selecting the most appropriate SkillPack based on task type. In the revised version, we will further include results demonstrating the effect of **router training** in these scenarios.
>
> **(3) Qualitative analysis of router behavior.**
> We add qualitative cases showing router behavior on inputs that mix biomedical, financial, and legal terms. For each example we report the top-1 SkillPack and the domain knowledge it encodes. Table 6 shows the router consistently selects the SkillPack aligned with the dominant semantic cue, and each SkillPack captures coherent domain expertise (e.g., dose–response reasoning in Biome), clarifying how mixed-domain queries are handled.
>
> Table 6: Qualitative Case Studies on SkillPack Knowledge.
>
> | Input Example                                                | Router (Top-1) | Knowledge Captured in SkillPack                              |
> | ------------------------------------------------------------ | ------------------------- | ------------------------------------------------------------ |
> | 1. “Does increasing the dose improve overall return under the current treatment policy?” | Biomed                    | Captures clinical reasoning such as dose–response relationships, treatment efficacy, and pharmacological effects. |
> | 2. “What regulatory constraints apply when reporting adverse outcomes that may affect financial liability?” | Law                       | Encodes legal concepts including regulatory compliance, reporting requirements, and liability interpretation. |
> | 3. “Evaluate whether the compound shows significant impact on long-term yield.” | Biomed                    | Stores biochemical and pharmacological knowledge regarding compound effects, biological impact, and experimental outcomes. |

---

> ### Author Response · Authors · 2025-11-25
> **Response to Reviewer vnrD (part 5/5)**
>
> > **Question 5: Presentation Issues and Missing Details**.
>
> **Reply**:  **5.1 Inconsistent Notation**: Thank you for the observation. We have standardized the base model notation across the paper (now consistently using θ_tgt) to avoid confusion.
>
> **5.2 Missing Quantization Details**：We appreciate the reviewer’s comment. In all our experiments, we use the **C4 validation split**(allenai/c4/en/c4-validation.00000-of-00008.json.gz) as the calibration set for GPTQ. This dataset is widely adopted in GPTQ-based quantization research, and we will clarify this detail for full reproducibility.
>
> **5.3 Formulation of the Router**: Thank you for pointing this out. We have moved the router description from the appendix into the main paper **Section 3.3** and provided clearer architectural details in **Appendix B.3**. In the *implicit model fusion* setting, the router is implemented as a lightweight feed-forward network with **four fully connected layers** (input → 4096-dim hidden → 1024-dim hidden → 256-dim hidden → 5-way output), equipped with GELU activations and LayerNorm. The **input features come from the Llama2-7B embedding head**, whose hidden dimension is 4096, and this 4096-dim representation is directly used as the router’s input. The router is trained using the same dataset as used for source-model distillation. We have also added the missing hyperparameters (batch size, learning rate, and training steps) in the revision. As described in the original submission, the supervision comes from the per-input losses of the five pairwise-distilled models, enabling the router to predict the most suitable source model for each input sample.
>
> **5.4 Claims about Parameter Size**: Our core contribution lies in ***dynamic fusion***, and existing dynamic fusion baselines require **far more than** 7B parameters. As shown in **Table 1** and **Figure 1**, compared with the pairwise-distillation + router approaches, our method achieves the best performance with **the smallest additional parameter cost**. Even though the final routed model is larger than the 7B base model, the performance is nearly **lossless** as using **multiple 7B fine-tuned** model.
>
> **5.5 Comparison in Continual Learning**: Thank you for the comment. **(1)** This concern has been addressed in the previous version of the paper, where we reported both the LLaMA3 sequential fine-tuning results and the individual fine-tuning results in **Appendix C.1** and **Table 11**.  **(2)** In the forget-free setting, we adopt an ideal task-type-based router to better illustrate the advantage of our router-based SkillPack composition. This design shows that our method effectively mitigates catastrophic forgetting—**a capability that prior approaches do not possess**. Importantly, implementing a simple router to determine the task type **introduces negligible overhead**, while the ability to fuse and compress multiple models is precisely the key strength of our framework.
>
> ---
> > ### **Motivation** behind our proposed GraftLLM is as follows:
>
> In practical applications, **multi-teacher distillation$^{[1]}$** often becomes extremely difficult to train due to severe **data heterogeneity**. FuseChat$^{[2]}$ and other methods attempted to address this by combining pair-wise distillation with parameter merging to mitigate incompatibility among heterogeneous LLMs. However, because parameter conflicts remain fundamentally unavoidable, **the improvements were limited**. Consequently, FuseChat-3.0$^{[3]}$ **abandoned parameter merging entirely and returned to pure multi-teacher distillation**.
>
> In contrast, our work **reintroduces dynamic (non-static) model merging** into the heterogeneous LLM setting, making model merging **feasible again** and **restoring confidence in the distillation + fusion paradigm**. Model merging has been widely adopted in recent years due to its simplicity, efficiency, and ease of use, and we hope that dynamic model merging can open a new direction for multi-LLM distillation.
>
> As modern LLMs rapidly expand toward multimodal and multi-capability scenarios, **data heterogeneity and cross-capability transfer** become increasingly challenging. GraftLLM provides a new dynamic fusion solution to address this challenge.
>
> ---
>
> ### **We thank the Reviewer again for the useful comments.** We have revised the manuscript according to the Reviewer’s suggestion and response to each comment provided in the Weakness and Question sections above. **We hope that our rebuttal aligns with the reviewer’s expectations**, and we hope that the Reviewer can consider possibly giving a higher rating. Thanks.

---

> ### Comment · Reviewer_vnrD · 2025-11-28
>
> I appreciate the author's detailed response and will raise my score.

---

> ### Author Response · Authors · 2025-11-28
>
> Thanks so much for raising your rating from 4 to 6. We are glad that we can solve all your questions and we are very much looking forward to further discussions on our work! Your comments really help improving our work.

---

### Official Review · Reviewer_Evyv · 2025-11-01

**Soundness:** 3
**Presentation:** 4
**Contribution:** 3
**Rating:** 6
**Confidence:** 4

**Summary:**

This paper identifies a critical challenge in cross-capability transfer for LLMs, i.e., existing methods like full-parameter knowledge distillation often cause catastrophic forgetting, while standard PEFT methods are ineffective at absorbing comprehensive knowledge from powerful source models. To address this, the authors propose GraftLLM, a novel grafting-based framework. The core idea is to distill and store capabilities from source models into modular, compressed "SkillPacks." These SkillPacks can then be attached to a target model to fuse new capabilities, preserving the target's original knowledge. The method employs a "module-aware adaptive compression" strategy to create these SkillPacks, aiming to reduce parameter conflicts and support "forget-free" continual learning and model fusion.

**Strengths:**

- I think the authors did a very good job of identifying a key gap: the lack of a method that both effectively absorbs deep knowledge from a source model (like distillation) and preserves the target model's inherent capabilities (which distillation often fails to do).
- The method introduced is sound and the reframing of model fusion as a modular composition problem is quite novel.
- A "forget-free" method for adding new, complex capabilities to a base model is highly sought after. This approach has clear significance for building specialized models on demand without the need for costly full-scale retraining.

**Weaknesses:**

- I think the baseline comparisons is a bit ambiguous. The abstract positions the work against distillation and PEFTs, and mentions FuseLLM/FuseChat for small models. However, it's unclear if GraftLLM is benchmarked against current SOTA model merging techniques for large models. These methods also aim to fuse capabilities and are a crucial point of comparison.
- The framework's practicality hinges on the cost of creating the SkillPacks. Can the authors please quantify the overhead of their method? What is the computational cost (e.g., GPU hours) to create a typical SkillPack? What is the inference-time impact (e.g., added latency, extra VRAM) of using a target model + 1 SkillPack?
- A key test for a fusion method is its ability to handle conflicting or highly distinct capabilities (e.g., fusing a "math" SkillPack and a "poetry" SkillPack). How does GraftLLM handle fusing multiple SkillPacks that may have conflicting parameter updates like this example?

**Questions:**

See weaknesses.

---

> ### Author Response · Authors · 2025-11-25
> **Response to Reviewer Evyv (part 1/4)**
>
> Thank you for your valuable comments. We will explain your concerns point by point.
>
> ---
>
> > **Weakness 1: Concerns about our Description of Prior Work.**
>
> **Reply**: We thank the reviewer for the comment. To clarify, our intention was **not** to suggest that FuseLLM/FuseChat target small models. These methods actually focus on **fusing multiple large-scale source models (>20B parameters) into a compact 7B model**. We will revise the abstract to make this distinction explicit.
>
> By contrast, much prior work on model merging mainly addresses **homogeneous architectures or smaller backbones (e.g., ViT)**, highlighting a gap in methods for **heterogeneous, large-model fusion**. Importantly, our experiments *do* benchmark against state-of-the-art large-model fusion approaches such as **InfiFusion**[1] and **Twin-Merging**[2]. We will clarify this further in the revision to avoid any confusion about our baselines.
>
> **Reference**: \
> [1] InfiFusion: A Unified Framework for Enhanced Cross-Model Reasoning via LLM Fusion; \
> [2]Twin-Merging: Dynamic Integration of Modular Expertise in Model Merging.

---

> ### Author Response · Authors · 2025-11-25
> **Response to Reviewer Evyv (part 2/4)**
>
> > **Weakness 2: Cost of Creating SkillPacks.**
>
> **Reply**: We provide a detailed resource analysis comparing SkillPack creation and deployment with representative fusion baselines. As summarized in Table 1 below, the **additional one-time compression stage** introduces minimal overhead, and SkillPacks maintain comparable **training costs** while incurring **<3% extra inference latency**, demonstrating that our approach is both **practical and scalable** for multi-capability LLM fusion.
>
> Table 1: Training and inference resource comparison between SkillPack and representative LLM fusion baselines.
>
> |                         Method                         | Model Size | Router Size | Training Cost (GPU h) | Compression Cost | Inference Overhead |
> | :----------------------------------------------------: | :--------: | :---------: | :-------------------: | :--------------: | :----------------: |
> |          Multi-Teacher Distillation (FuseLLM)          |     7B     |      –      |         132 h         |        –         |       1.00×        |
> |  Pairwise Distillation + Parameter Merging (FuseChat)  |     7B     |      –      |         132 h         |      4 min       |       1.00×        |
> | Pairwise Distillation + Dynamic Merging (Twin-Merging) |    21B     |     21M     |     132 h + 8 min     |      9 min       |       1.03×        |
> |    **Pairwise Distillation + SkillPack (Ours)**    |    9.2B    |     21M     |     132 h + 8 min     |       163 min        |       1.03×        |

---

> ### Author Response · Authors · 2025-11-25
> **Response to Reviewer Evyv (part 3/4)**
>
> > **Weakness 3: Results on fusion domains with much larger intrinsic divergence.**
>
> **Reply**: Our method is explicitly designed to **decouple conflicting task behaviors into separate SkillPacks**, preventing cross-task interference and enabling near-lossless capability fusion—even across highly distinct domains.
>
> To demonstrate this property, we conducted an additional experiment on domains with **much larger intrinsic divergence** (finance, law, and biomedicine), following the evaluation protocol of the ICLR 2024 paper *“Adapting Large Language Models to Domains via Reading Comprehension.”*
>
> We first evaluate models individually fine-tuned on each domain. As shown in the Table 2 below, a model fine-tuned on one domain suffers substantial degradation on the others, sometimes performing worse than the base model. This confirms that traditional parameter-merging approaches cannot resolve strong **negative interference across domains**.
>
> **In contrast, our SkillPack-based fusion effectively isolates domain-specific updates and recombines them with minimal interference, achieving near-lossless multi-domain performance.**
>
> Table 2: Performance comparison between GraftLLM and representative LLM fusion baselines, evaluated across three domains: Biomedicine, Finance, and Law.
>
> |             Methods              | Params   | Biomedicine | Finance  |   Law    | Average        |
> | :------------------------------: | -------- | :---------: | :------: | :------: | -------------- |
> |             LLaMA-7B             | 7B       |    44.2     |   58.6   |   34.2   | 45.7           |
> |    AdaptLLM-7B [1]\[ICLR24\]     | **3*7B** |    47.3     |   63.4   |   38.5   | 49.7_(100%)    |
> |           LLaMA-7B-Bio           | 7B       |  **47.3**   |   57.9   |   34.5   | 46.6           |
> |         LLaMA-7B-Finace          | 7B       |    43.7     | **63.4** |   34.0   | 57.0           |
> |           LLaMA-7B-Law           | 7B       |    44.1     |   58.2   | **38.5** | 46.9           |
> | AdaptLLM-7B + Parameter Merging  | 7B       |    45.6     |   60.1   |   36.3   | 47.3_(95%)     |
> |           Twin-Merging           | 15.3B    |    45.9     |   61.3   |   36.7   | 47.9_(96%)     |
> | LLaMA-7B + Router + 3*SkillPacks | **9.1B** |    47.2     |   63.4   |   38.2   | **49.6_(99%)** |

---

> ### Author Response · Authors · 2025-11-25
> **Response to Reviewer Evyv (part 4/4)**
>
> > **Router Behavior and Qualitative Case Studies on SkillPack Knowledge**.
>
> **(1) When tasks are highly similar (Explicit and Implicit LLM Fusion settings).**
> When tasks share close semantic characteristics, distinctions between experts become subtle, making **top-1 routing inherently ambiguous**. In these scenarios, **ensembling multiple SkillPacks** (e.g., top-2 routing) can provide **a noticeable performance gain** by aggregating complementary updates from closely related experts. However, this comes at the cost of increased inference latency and memory usage. For fairness and efficiency, we use only the top-1 SkillPack during inference, while noting that ensembling offers an optional boost in accuracy for such cases.
>
> **(2) When tasks are clearly differentiated (Finance–Bio–Law LLM Fusion settings).**
> When tasks are more distinct, the router can easily learn the optimal mapping from input patterns to source capabilities. In these settings, **top-1 routing is sufficient**, as the router naturally emphasizes the most relevant expert model. Cross-expert synergy plays a smaller role because the knowledge from different tasks does not strongly reinforce each other. This behavior is consistent with the trends observed in **Table 2 and Section 6.3**.
>
> Table 3: Analysis of router behavior and multi-SkillPack ensembling across three fusion settings.
>
> |          Methods          | Explicit LLM Fusion (*AlpacaEval 2.0*) | Implicit LLM Fusion (*10 benchmark*) | Fusion Finance+Law+Bio LLM | Inference Time |
> | :-----------------------: | :------------------------------------: | :----------------------------------: | :------------------------: | -------------- |
> |         FuseChat          |             11.63 / 16.89              |                 55.2                 |           47.33            | 1x             |
> | Grafting Top 1 SkillPack  |             16.56 / 25.42              |                 56.0                 |           49.62            | 1.03x          |
> | Grafting Top 2 SkillPacks |  17.33 (**+0.8**) / 26.28 (**+1.8**)   |           56.3 **(+0.4)**            |   **49.64** **(+0.02)**    | 2.03x          |
> | Grafting Top 3 SkillPacks |  **17.49** (+0.2) / **26.72** (+0.4)   |           **56.4** (+0.1)            |       49.63 (-0.01)        | 3.03x          |
>
> ---
>
> **We thank the Reviewer again for the useful comments.** We have revised the manuscript according to the Reviewer’s suggestion and response to each comment provided in the Weakness section above. **We hope that our rebuttal aligns with the reviewer’s expectations**, and we hope that the Reviewer can consider possibly giving a higher rating. Thanks.

---

> > ### Comment · Reviewer_Evyv · 2025-11-27
> > **Response by Reviewer Evyv**
> >
> > I thank the authors for the detailed response. All my queries have been resolved, and I would like to maintain my score.

---

> > > ### Author Response · Authors · 2025-11-28
> > >
> > > Thanks so much for maintaining your positive rating of our work. We are so glad that all your problems are solved and we are very much looking forward to further discussions on our work!

---

### Author Response · Authors · 2025-12-02
**Summary of Rebuttal Updates & Discussion Status**

Dear Area Chair and Program Chair,

We sincerely **appreciate your time and attention** to our submission, particularly in light of the recent administrative challenges. To facilitate your review of the rebuttal exchanges and our revised manuscript, we provide a concise summary of the current reviewer status and the key improvements made during the discussion period.

**1. Consistent Reviewer Agreement and Rating Updates.**

We have received responses from all five reviewers, and **all of them** indicated that their concerns have been addressed, with no new issues raised. We are pleased to report that three reviewers **explicitly acknowledged the improvements and increased their ratings**, as shown in the table below. Importantly, all of these rating updates occurred **before** the reported data leak incident (Nov 27, AOE time), except for reviewer vnrD, whose update came only a few hours after the incident. Thus, the score increases were made based on our rebuttal and the fact that we had **indeed resolved the reviewers’ concerns**.

Additionally, we believe that reviewer **HvCk** demonstrates a limited understanding of fundamental concepts in **LLM** and **NLP**, along with **limited English comprehension**, which led to numerous and fundamental **misunderstandings** of our work. The review appears to be of exceptionally low quality and anomalous in nature. We respectfully request that the AC/PC consider discounting this reviewer’s rating to ensure a fair and reasonable final evaluation of our research work.

|         Reviewers          |  Evyv  |  vnrD  |  HvCk  |  zW7i  |  ZGH4  | Average |
| :------------------------: | :----: | :----: | :----: | :----: | :----: | :-----: |
|       Initial Rating       | **6**  |   4    |   2    |   4    | **6**  |   4.4   |
|         Confidence         |   4    |   4    |   3    |   3    | **5**  |   3.6   |
|         Reply time         | Nov 26 | Nov 27 | Nov 27 | Nov 25 | Nov 25 |         |
|      **Final Rating**      |   6    |   6    |   2    |   6    | **8**  |   5.6   |
| Proposed Rating (w/o HvCk) |   6    |   6    |   -    |   6    | **8**  | **6.5** |

**2. Current Reviewer Discussion Status.**

- **Reviewer Evyv (Confidnece 4, Rating: 6):** All queries have been resolved.
- **Reviewer vnrD (Confidnece 4, Rating: 4 $\to$ 6):** All questions have been resolved and no further concerns.
- **Reviewer HvCk (Confidnece 3, Rating: 2 ):** All comments have been addressed.
- **Reviewer zW7i (Confidnece 3, Rating: 4 $\to$ 6):** All questions have been addressed.
- **Reviewer ZGH4 (Confidnece 5, Rating: 6 $\to$ 8):** All concerns have been addressed.

**3. Detailed Revisions.**

To comprehensively address all reviewer feedback, we have uploaded a revised PDF. The major updates include:

- **Introduction:** Added a definition of *cross-capability transfer* in **Section 1** (addressing **Reviewer HvCk**).
- **Methodology:** Clarified router-related descriptions in **Section 3.3** (addressing **Reviewers vnrD and ZGH4**).
- **New Experiment:** Added results on fusing LLMs with highly distinct capabilities in **Section 5.4** (addressing **Reviewer Evyv**).
- **Router Behavior:** Included an extended analysis of router behavior in **Section 6.3** (addressing **Reviewers zW7i and vnrD**).
- **Inference Resource Usage:** Reported runtime and computational complexity compared with prior methods in **Appendix F.3** (addressing **Reviewers ZGH4 and zW7i**).

We hope this summary facilitates your assessment. We remain fully available for any further questions.

Best regards,

The Authors

---

### Meta-Review · Area_Chair_FzoA · 2026-01-15

**Summary:**

Acceptance is recommended. The paper proposes GraftLLM, a framework for cross-capability transfer in LLMs using modular "SkillPacks" and a router-based fusion mechanism. The majority of reviewers (Scores: 6, 6, 6, 8) found the approach innovative and effective for heterogeneous model fusion, enabling "forget-free" learning. The single negative score (2) appears to be an outlier based on misunderstandings of fundamental concepts (e.g., "heterogeneous models") which were clarified in the rebuttal.

**Reviewer Concerns:**

The authors successfully addressed the substantive concerns raised during the review process:

Baselines: The comparison was expanded to include state-of-the-art large model fusion methods (e.g., InfiFusion, Twin-Merging) and challenging cross-domain settings (Law/Finance/Bio), addressing concerns from a reviewer.

Router Mechanism: The formulation and behavior of the router were clarified, including an analysis of top-k routing and training details, resolving confusion for reviewers.

Efficiency: A detailed breakdown of training and inference costs (showing improved latency overhead) was provided to address practicality concerns.

**Reviewer Scores:**

Scores improved significantly following the rebuttal. The dissenting reviewer (HvCk) maintained a score of 2 despite their comments being addressed; given the potential lack of domain expertise noted in the author's response, this score should be weighed less heavily against the strong consensus of the other four reviewers.

---

### Decision · Program_Chairs · 2026-01-26

Accept (Poster)